# Changes in the Population Genetic Structure of Captive Forest Musk Deer *(Moschus berezovskii)* with the Increasing Number of Generation under Closed Breeding Conditions

**DOI:** 10.3390/ani10020255

**Published:** 2020-02-05

**Authors:** Yonghua Cai, Jiandong Yang, Jianming Wang, Ying Yang, Wenlong Fu, Chengli Zheng, Jianguo Cheng, Yutian Zeng, Yan Zhang, Ling Xu, Yan Ren, Chuanzhi Lu, Ming Zhang

**Affiliations:** 1Sichuan Institute of Musk Deer Breeding, Dujiangyan, Chengdu 611845, China; Wangjm@scyss.com.cn (J.W.); Yangying@scyss.com.cn (Y.Y.); zengchl@scyss.com.cn (C.Z.); Chengjg@scyss.com.cn (J.C.); 2College of Animal Science and Technology, Sichuan Agricultural University, Chengdu campus, Wenjiang 611130, China; yangjd@sicau.edu.cn (J.Y.); fuwl@scyss.com.cn (W.F.); xuling1114684263@163.com (L.X.); ryan961231@163.com (Y.R.); m17381580996@163.com (C.L.)

**Keywords:** forest musk deer, microsatellite DNA, genetic diversity, population genetic structure

## Abstract

**Simple Summary:**

Forest musk deer is an endangered species. Musk produced by male musk deer is precious natural flavor and an important ingredient of traditional Chinese medicines. Forest musk deer is a territorial animal, and it is very difficult for captive forest musk deer to record the accurate pedigree using the “Breeding Management System of Rotated Mating” (BMS-RM). In the study, we examined the genetic structure of captive forest musk deer population in *Barkam* center, and the changes in allele diversity, heterozygosity and inbreeding coefficient with increasing generations. The results show that the population has high genetic diversity, but the genetic structure of different generations in the population shows that the extent of inbreeding is slowly increasing with increasing number of generations. This suggests that the BMS-RM is effective for maintaining genetic structure diversity of population, but captive forest musk deer in *Barkam* center still face the risk of inbreeding increasing. So, it is necessary to optimize the BMS-RM of captive forest musk deer or introduce male forest musk deer from an unrelated population. This study revealed the status of genetic diversity of captive forest musk deer in *Barkam* center and the current risks of inbreeding for providing an important understanding to the application of BMS-RM.

**Abstract:**

We investigated the genetic diversity of the population of captive forest musk deer (*Moschus berezovskii*) in *Barkam* Musk Deer Breeding Centre using twelve microsatellite markers, and then analyzed the change in genetic structure of successive generation groups from the population. The data provide a new understanding for the evaluation and usage of the breeding management system. Microsatellite marker analysis detected 141 alleles with an average of 11.75 alleles for each marker. The average expected heterozygosity (H_E_) was 0.731. Performing an *F*-statistical analysis on the data showed that the genetic diversity of population decreased, and the inbreeding coefficient significant increased with the increase of generation, and F_IS_ of the 1st generation is significantly lower than that of the second to fifth generation (*p* < 0.01). The result suggested that the captive population was facing the pressure of inbreeding (F_IS_ = 0.115) and the subsequent loss of genetic diversity. Therefore, it is necessary to improve the breeding management system of the captive population by preventing close relatives from mating or inducing new individuals from the exotic population.

## 1. Introduction

The forest musk deer (*Moschus berezovskii*) is an Asian ungulate that is listed as an endangered species by the International Union for Conservation of Nature (IUCN) [1]. It is included in the Convention on International Trade in Endangered Species of Wild Fauna and Flora (CITES) Appendix II and protected wild animal in China [2]. The forest musk deer widely distributed in many provinces in China and in the northern part of Vietnam [3]. Musk produced by male musk deer has high value, and it is used as a precious natural flavor and an important ingredient of traditional Chinese medicines. Therefore, wild male forest musk deer were illegally hunted for collecting musk. From the late 1960s to the late 1990s, the number of wild musk deer dropped sharply from one million animals to about two hundred thousand [4,5].

The breeding of captive forest musk deer originated in 1958 in China. The early breeding technique was mainly to expand the population, explore the method of musk collection and improve the husbandry conditions of timid forestry musk deer [4]. After the non-invasive technique for musk collection was successfully developed and the Chinese government commenced the program of captive forest musk deer breeding for musk harvesting in 1958 [6], some breeding centers of musk deer, such as *Barkam, Miyaluo, Dujianyan, Fengxian, Yaan,* have been established to relieve the severe hunting pressure to wild populations and provide markets with legal musk obtained from captive animals for traditional Chinese medicine [7]. However, captive populations are facing the problem of inbreeding, the loss of genetic diversity and the increased diseases [8]. Also, high genetic diversity of captive forest musk deer is the basis of artificial breeding for increasing musk yield [9]. Among these centers, *Barkam* center was the first to be established [7], and consequently, studies on the genetic diversity of captive forest musk deer in *Barkam* center are of great importance for maintaining genetic diversity, expanding the population, and assessing the breeding management system of the captive forest musk deer. Forest musk deer is a solitary, territorial species that is easily frightened [4]. If male and female forest musk deer randomly mate within a large population in a center, it is very difficult to record the accurate pedigree of offspring under captive conditions [5]. To avoid inbreeding, record relatively clear pedigrees of the offspring, and implement breeding management, a large population of captive musk deer was usually divided into many breeding groups in the breeding center, and the males and females within one breeding group mate randomly during the breeding season. In the next breeding season, the male musk deer were transferred to the other breeding group within the population. The population didn’t induce forest musk deer form other breeding centers as a closed breeding population during the certain period. This breeding management system of rotation mating (BMS-RM) was widely applied in captive forest musk deer in China [7]. However, whether the BMS-RM can maintain the stability of genetic structure and avoid inbreeding in a closed breeding population has not been reported.

Microsatellite DNA, also known as simple sequence repeats (SSRs) or short tandem repeats (STRs), consists of 1–6 core sequences repeated in series. With good stability, wide distribution, dominant inheritance, and many other advantages, SSRs provide excellent markers for measuring genetic diversity [10,11] and looking for markers of valuable traits in a species such as the forest musk deer [12,13,14,15,16]. Microsatellites are also useful for determining the potential for hybridization [17,18], for quantitative trait locus (QTL) mapping and trait linkage analysis [19,20], and as the basis for molecular marker selection in animal breeding.

Other molecular genetic methods have been applied to the study of genetic diversity in forest musk deer populations. The amplified fragment length polymorphism (AFLP) marker technique was used to compare the genetic diversity of two populations of captive forest musk deer in *Baisha* of Sichuan Province and *Jinfeng* Mountains by Zhao et al. [21]. Peng et al. [22,23] and Feng et al. [24] analyzed genetic diversity of the populations of captive forest musk deer in *Sichuan* and *Shaanxi* Provinces, China, by comparing their mitochondrial DNA-loop region’s sequence structure. Since then, more microsatellite loci have been found in forest musk deer [3,17,25,26]. Guan et al. [27] profiled the genetic structure of three captive forest musk deer populations in *Miyaluo*, *Jinfeng* Mountain and *Barkam* by using eleven microsatellite loci. The results revealed 142 different alleles in 121 individuals. The expected heterozygosity (H_E_) in the three populations was 0.552, 0.899, and 0.894, respectively, which indicated the three populations had high genetic diversity. The genetic diversity of two musk deer populations in *Baisha* and *Jinfeng* Mountain of *Sichuan* Province was calculated using 15 microsatellites and 22 AFLP primers [21] and a high genetic diversity was found in the both populations. Huang et al. compared young to adult forest musk deer in *Miyaluo* using seven microsatellite markers, and the genetic diversity of the younger musk deer was higher than that of the adult group [28].

In this study, our objective was to evaluate genetic structure of the population at the musk deer farm of Sichuan Institute of Musk Deer Breeding in *Barkam* and examine the change in genetic structure of successive generations based on the polymorphisms at twelve microsatellite loci. We then also compared the distribution of private alleles and inbreeding coefficients of different generations and provided a scientific reference for the inbreeding control and genetic diversity maintaining of BMS-RM. If there is a high genetic diversity in the closed population, this implies the present BMS-RM is effective in maintaining the stability of genetic structure. If the inbreeding coefficient does not significantly increase with the increasing number of generations, this impliesy the present BMS-RM is effective in avoiding inbreeding. Otherwise, the BMS-RM will need to be improved in the future.

## 2. Materials and Methods

### 2.1. Animals and Breeding Management System

The forest musk deer were housed and raised in the farm of captive forest musk in *Songgang* town, *Barkam*, *Sichuan* Province, and there are more than 300 forest musk deer in *Barkam* center for genetic protection and musk production. The population originated from 18 wild forest musk deer in 1958, and some wild individuals were added to the population until 1980. The generations of these individuals were recorded, and their blood samples were collected at the time of musk collection or vaccination. These blood samples have been collected continuously for nearly 20 years, and were kept at −80 °C. All operations strictly obeyed Wildlife Protection Acts and Regulations of China, and were also approved by the Wildlife Protection Committee of Musk Deer Research Institute (MDRI-2009-02), and the Ethics Committee of Use of Endangered Wild Animals in Research and Teaching (2018YSZH0019).

Forest musk deer is seasonal breeding animal, and the breeding season lasts from October until the next February. During the breeding season, 17 forest musk deer (Male: 2; Female: 15) in a breeding group are housed in the same shed, and the male and female forest musk deer mate freely. The two male forest musk deer are transferred to another shed for mating with another 15 females in the next breeding season. The offspring of 3-year-old male and 2.5-year-old female musk deer begin to be used for reproduction in the next generation [7]. The number of male musk deer which is selected for breeding is depend on the number of female musk deer in the offspring population (according to female:male = 15:2), and the rank of musk yield of the male offsprings, and other male forest musk deer in offspring population are not used for breeding but only for musk producing. All female descendants are used for breeding. The breeding management system of rotated mating (BMS-RM) could avoid the mating of the same male and female musk deer within at least five years (the number of breeding group at least is more than 5). The breeding of all musk deer in *Barkam* center is strictly adheres to the rules of BMS-RM, and all individuals avoid sib-pair and parent-offspring mating (Figure 1).

### 2.2. Blood Collection and DNA Extraction

We chose 238 forest musk deer blood samples with generational record. All individuals (Female: 142, Male: 96) were from 5 successive generations (1st generation: 32; 2nd generation: 48; 3rd generation: 55; 4th generation: 52, and 5th generation: 51). The blood samples were preserved at −80 °C after addition of citric acid-EDTA as anticoagulant. Genomic DNA was extracted using the phenol-chloroform method [29] and stored at −20 °C.

### 2.3. PCR Amplification of Microsatellite Sequences and Identification of Alleles and Genotype

Twelve microsatellite loci with a high degree of polymorphism and stable amplification were selected for this study. The primers, which were synthesized by Shanghai Biotechnology Ltd. Co. (Shanghai, China), are shown in Table 1 [3,17]. The volume of the PCR reaction was 15 μL: 1.5 μL of 10x buffer, 1 μL of Mg^2+^ (25 mmol/L), 1 μL dNTPs (2 mmol/L), 1 μL of each upstream and downstream primer (10 μmol/L), 1 μL of template DNA, 1 U of Taq DNA polymerase (Promega, Madison, WI, USA), and 7.5 μL ddH_2_O. PCR amplifications were performed using a Perkin Elmer 9700 thermocycler (Perkin Elmer, Waltham, MA, USA). The PCR amplification conditions were: 95 °C pre-denaturation for 5 min, 94 °C denaturation for 30 s, annealing at a specific temperature (Table 1) for 50 s, 72 °C extension for 50 s, 35 cycles; 72 °C extension for 10 min. Aliquots of 5 μL of each PCR amplification product were analyzed by capillary electrophoresis using an automated ABI PRISM™ 3700 Genetic Analyzer (Applied Biosystem Inc., Waltham, MA, USA), and the products size were identified using Gene Mapper V4.0 (Applied Biosystem Inc., Waltham, MA, USA).

### 2.4. Data Analysis

The genetic diversity parameters of the population, including the number of alleles (A), observed heterozygosity (H_O_), expected heterozygosity (H_E_), and polymorphism information content (PIC) were determined using CEVUS 3.0 software [25]. Linkage disequilibrium (LD) and Hardy-Weinberg genetic equilibrium (HWE) were calculated with GENEPOP 3.4 software [26]. The gene richness (AR), Wright’s inbreeding coefficient (F_IS_), and population differentiation coefficient (FST) of each microsatellite locus were calculated with the FSTAT 2.9.3 software package. F_IS_ of all generation in 12 microsatellite loci were compared by *F*-test. H_E_ of different generation was compared by the Wilcoxon nonparametric test using GENEPOP 3.4 [26]. The expected F_IS_ increment (ΔFIS_E1_) of the population of random mating (238 individuals) was obtained using the formula: (1)$$\Delta \mathrm{FIS}E_{1}=\frac{N_{m}\;+\;N_{f}}{8N_{m}N_{f}}$$
where *N_m_* is the number of male individuals, while *N_f_* is the number of female individuals. The expected FIS increment (ΔFIS_E2_) of the breeding group (2 male and 15 female) was obtained by the formula: (2)$$\Delta \mathrm{FIS}E_{2}=\frac{32N_{m}N_{f}}{N_{m}\;+\;3N_{f}}$$
where *N_m_* = 2 and *N_f_* = 15. The actual FIS increment (ΔFIS_A_) in the breeding management system was obtained according the formula: (3)$$\Delta {\mathrm{FIS}}_{A}=\frac{\sum_{i}^{n-1}\left(F{\mathrm{IS}}_{i}-FIS_{i-1}\right)}{n-1}$$
where *i* is generation and *i* = 1, 2, 3, 4 and 5 in the study [9].

## 3. Results 

### 3.1. Genetic Diversity Analysis of Musk Deer Population

Our captive musk deer population had a high degree of genetic diversity (Table 2). 238 individuals were genotyped in the Mb118H, Mb116H, Mb39, Mb38 and Mb33 loci, but only 86 individuals were identified genotypically in the Mb40 locus, and 94 individuals were identified genotypically in the Mb41. We found 141 alleles at 12 microsatellite loci from the sampling of 238 animals. The average number of alleles and the average polymorphism information content (PIC) was 11.8, and 0.699, respectively, and the expected heterozygosity (H_E_) was 0.731. The population had a large number of rare alleles. We identified 78 rare alleles at the 12 microsatellite loci with an average of 6.5 alleles per locus. In addition, the F_IS_ value also demonstrated a moderate degree of inbreeding risk in the population, since the inbreeding coefficient was 0.115, which is greater than 0.05 [26]. Four microsatellite loci (Mb102C, Mb39, Mb34, Mb18) significantly deviated from Hardy-Weinberg equilibrium (*p* < 0.01), while the remaining eight were in balance. After a correction using the Bonferroni method, the results of linkage disequilibrium analysis showed that the twelve microsatellite loci did not have any linkage between each other.

### 3.2. Comparison of Genetic Diversity among Different Generations in the Population

Female forest musk deer are seasonal breeding animals, and produce one generation about 3 or 4 years [7]. The allelic diversity and genetic parameters of five different generations in the population of captive forest musk deer were determined (Table 3). The alleles at each generation first increased and then decreased. From the 1st to the 5th generation, the average number of alleles increased from 8.25 to 9.25, and the allele abundance increased from 6.177 to 6.325. There was a largest number of private alleles (P_R_ = 6) in the 1st generation compared to other subsequent generations, and the number of private alleles continuously decreased from the 1st to the 4th generation because some male musk deer offspring didn’t participate in the breeding. The number of private alleles increased from the 4th to the 5th generation because rotation mating induced new male musk deer from other shed. The number of private alleles might vary up and down around a constant value, if more generations were examined because the breeding obeyed the BMS-RM in the closed population. At the same time, the number of rare alleles (less than 5% of the population) was highest in the 4th generation group (R_A_ = 49). However, the risk of loss of private and rare allele was always present, as some male individuals were not involved in breeding.

Five generations showed high heterozygosity (Table 4). However, the observed heterozygosity (H_O_) and the expected heterozygosity (H_E_) gradually decreased from the 1st to the 5th generation. The H_O_ and the H_E_ were, respectively, 0.743 and 0.742 in the 1st generation, and 0.632 and 0.731 in the 5th generation. The Wilcoxon nonparametric test showed that there was no significant difference in H_E_ between the 1st and the 5th generation (*p* > 0.05). The population inbreeding coefficient (F_IS_) showed a gradual increase from the 1st generation to the 5th generation. The minimum at 1st generation was 0.001, and the degree of inbreeding in the 2nd (F_IS_ = 0.134), the 3rd (F_IS_ = 0.112), the 4th (F_IS_ = 0.137) and 5th (F_IS_ = 0.139) generation extremely significantly increased compared with the 1st generation (*p* < 0.01). The expected FIS increment (ΔFIS_E1_) of the population of random mating was 0.005, and the expected FIS increment (ΔFIS_E2_) of the breeding group was 0.050, while actual FIS increment (ΔFIS_A_) in the breeding group was 0.034. The result suggested the BMS-RM was effective to maintain diversity in a captive population. However, the inbreeding of close relatives must be prevented to control the increase of inbreeding coefficient.

## 4. Discussion

### 4.1. Genetic Characteristics of Captive Musk Deer Population

Microsatellites are molecular markers with a high degree of polymorphism, good stability and wide availability in mammals. They are useful molecular markers in the field of population genetic analysis and have been widely used in genetic protection of wildlife [12,23,30,31,32,33]. In the analysis of microsatellite data, the number of microsatellite loci (A), the heterozygosity (H_E_), and polymorphic information content (PIC) are the important indicators of the level of diversity. Botstein et al. proposed a measure of PIC variability, with PIC < 0.25 being a low level, 0.25 < PIC < 0.5 moderate, and PIC > 0.5 a high level [34]. Zhou et al. identified eight highly polymorphic microsatellite loci that could be used with the improved genomic library of captive or wild forest musk deer to study the genetic diversity [18]. In our study, twelve microsatellite loci were used to detect polymorphisms in the population of captive musk deer, and the amplified results were stable. The average number of alleles was 11.75, and the average PIC was 0.699, which indicated a high genetic diversity according to Botstein’s criteria. The level of genetic diversity in our study implied the effectiveness of inbreeding control using BMS-RM in *Barkam* center. Xia et al. studied genetic diversity in a population of 48 captive forest musk deer using six microsatellite loci and found A = 8.67, H_E_ = 0.842 and PIC = 0.811 [3]. Zhao et al. [25] used 15 microsatellite loci to screen 31 captive forest musk deer in *Dujiangyan* center of *Sichuan* Province and found A = 9.4, H_E_ = 0.820, and PIC = 0.778. Guan et al. [27] used 11 microsatellite loci to study three populations of captive forest musk deer in *Miyaluo* center, *Jinfeng* Mountains center and *Barkam* center, and found A = 12.91, H_E_ = 0.899, and PIC = 0.884. Huang et al. [28] used 7 microsatellite loci to genotype captive forest musk deer from three groups in *Barkam* center, and found A = 24, H_E_ = 0.854, and PIC = 0.837. Our results (A = 11.75; H_E_ = 0.731; PIC = 0.670) were similar to theirs, confirming the high genetic diversity of captive musk deer in China. Other wild deer populations have lower diversity: red deer(*Cervus elaphus*, A = 5.78–7.28; H_E_ = 0.638–0.689) [35]; sika deer (*Cervus nippon pseudaxis*, A = 5.70; H_E_ = 0.600) [36], and white-tailed deer, (*Odocoileus virginianus*, A = 5.90–9.20; H_E_ = 0.670–0.730) [37]. The results suggested the genetic diversity of captive forest musk deer maintained a high level compared with other wild cervidae animals (red deer, sika deer and white-tailed deer). The possible reasons for this are to take measures of inbreeding control in the population of captive forest musk deer, while the population of wild deer live in fragmentation habitat lead to a higher degree of inbreeding.

The AFLP was first applied to measure genetic diversity of two populations of captive musk deer populations in *Baisha* of Sichuan and at a reserve in *Jinfeng* Mountains, and both populations had high genetic diversity [21,27], but the population in *Jinfeng* Mountain was higher than that in *Baisha* [21]. Peng et al. detected 27 haplotypes by analyzing the mitochondrial DNA (mtDNA) D-loop sequence of three captive musk deer herds in *Sichuan* Province [23], and Feng et al. also studied the mtDNA D-loop sequence of one captive population and three wild groups in *Shaanxi* Province, China [24]. They found that the mtDNA D-loop sequence of musk deer in *Shaanxi* Province showed considerable variation [23,24]. The genetic differentiation of both wild and captive groups was small, but there was a high degree of exchange of genetic material [24]. These results of previous studies showed that the genetic diversity of captive forest musk deer in China has maintained a high level, which reflected the effectiveness of inbreeding control in the artificial breeding management system of captive forest musk deer. Our results showed that the genetic diversity of captive forest musk deer in *Barkam* center was maintained a high level under closed breeding condition, and it implied that the BMS-RM was effective in inbreeding controlling.

### 4.2. Analysis of Genetic Diversity in Populations at Different Generations

Protecting the gene pool of a population requires preserving its genetic diversity. DNA replication, non-random mating, selection and drift, and other factors all tend to lead to the loss of diversity [38]. In our study we found that the genetic diversity at the 1st generation was the highest (H_E_ = 0.742), and Wright’s inbreeding coefficient at the 1st generation was the lowest (F_IS_ = 0.001) among the other generations. Consistent with our findings, Huang et al. reported that the genetic diversity of adult captive population was higher than that of younger population [28]. Captive musk deer population in *Barkam* center is at the risk of genetic diversity loss judged form the increase of inbreeding coefficient from the 1st generation to the 5th generation. On the contrary, our result also showed that the BMS-RM also maintained genetic diversity, even improved the genetic structure of the captive musk deer population judged from the average number of alleles and the allele richness: the average number of alleles increased from 8.25 at the 1st generation to 9.25 at the 5th generation, and the allele richness increased from 6.177 at the 1st generation to 6.325 at the 5th generation. New private alleles appeared at the Mb102C (two new private alleles) and Mb34 (one new private allele) locus. Conversely, a private allele at Mb116H locus was detected in the 1st generation, but was not detected in the 4th, 3rd and 2nd generation. The loss of private allele on Mb116H locus may be due to that some male offspring individuals didn’t involve in mating. Huang et al. came to a similar conclusion in their study [28]. Maybe, different parameters of judge genetic diversity reflect the long-term or short-term dynamic changes. However, the loss of genetic diversity is inevitable in the closed breeding population with the increasing number of generations in the future, and the BMS-RM can only reduce the speed of genetic diversity loss and inbreeding increase, or improve genetic structure in a short period. Therefore, the effective strategies are to control the inbreeding coefficient and to exchange individuals among different breeding centers for maintaining genetic diversity, and to induce wild forest musk deer for improving genetic diversity.

It is worth noting that the newly added genes in generation groups were mostly rare alleles with less than 5% of gene frequency. Our study showed that the 5th generation group had more private alleles (P_R_ = 5) and a higher number of alleles overall (A = 49). These rare alleles only were found in a few individuals, and they commonly disappeared from the gene pool. One founder of wild forest musk deer should be sired a sufficient number of offspring in order to prevent the pedigree bottleneck effect. Ballou’s [32] study suggested that effectively increase the genetic contribution of new wild founders to a population as well as increase the reproductive life span of existing founders and their close descendants acted to reduce genetic drift and inbreeding effects in the population and thereby minimize the loss of genetic diversity, therefore each of founder should produce at least seven pairs of descendants. The number of wild forest musk deer has been so severely reduced that now the State Forestry Administration has strict orders to arrest and prosecute any persons harming these wild musk deer. This is a national action to protect the founders. The captive breeding program is our main hope for species protection of forest musk deer, maintaining the genetic diversity, and breeding more captive forest musk deer to provide enough musk for traditional Chinese medicine.

Judging from the inbreeding coefficient increment (ΔFIS), the BMS-RM decreased the risk of inbreeding compared with the mating system of random mating within the breeding group, and with the non-rotation mating system of male musk deer across different generations, but it also increased the degree of inbreeding compared with the random mating system in a large population. It is difficult to randomly mate in a large population of captive individuals. Indeed, random mating of large population is easy to cause the repeated mating and missed mating of female musk deer [5]. Therefore, the BMS-RM is widely used [21]. In order to decrease the risk of inbreeding and ΔFIS, the number of individual within the population (*N*), the appropriate number of individuals within breeding group (*N_BG_*), the number of breeding groups within the large population ($\frac{N}{N_{BG}}$), the number of male musk deer within breeding group (*N_m_*) and the exchange of male musk deer across different generations were considered as important factors. It’s necessary to further clarify the proper size of *N*, *N_BG_*, and *N_m_* in order to achieve the minimum inbreeding increment using the BMS-RM under closed breeding conditions, and we advocate the exchange of individuals among different breeding centers, and to occasionally induce wild forest musk deer as the complementary strategy of BMS-RM in China.

## 5. Conclusions

In summary, the finding of our present study is that the genetic diversity of captive forest musk deer in *Barkam* center maintained a high level. The breeding management system of rotated mating (BMS-RM) is effective in maintaining high genetic diversity, but the genetic diversity of population slowly decreased and inbreeding coefficient slowly increased with increasing number of generations, which implied the captive population is facing the pressure of inbreeding and subsequent loss of genetic diversity. Therefore, it is necessary to improve the breeding management system or develop new breeding management system of the captive population in the future.

## Figures and Tables

**Figure 1 animals-10-00255-f001:**
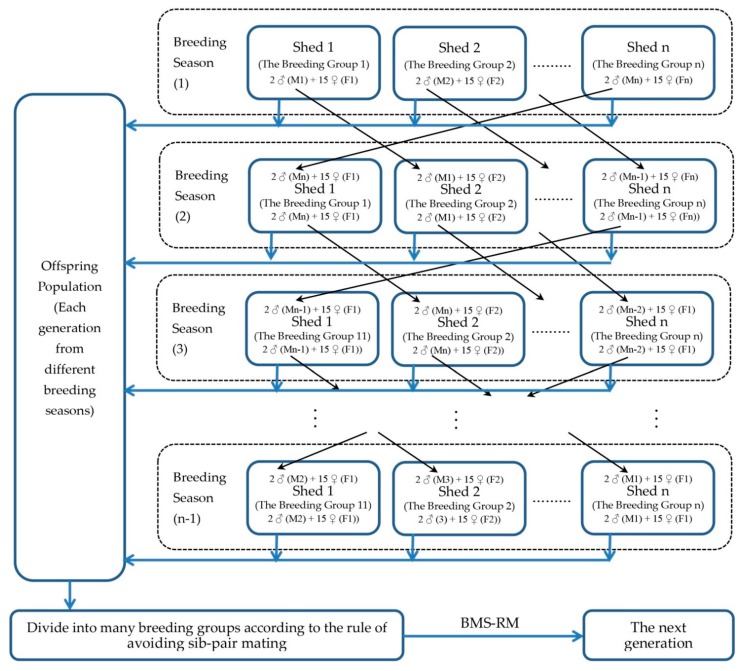
The schematic diagram of BMS-RM.

**Table 1 animals-10-00255-t001:** The primer sequences of 12 microsatellite loci and annealing temperatures of PCR amplification.

Locus ID	Repeated Sequence	Primer Sequence (5′→3′) [25,26]	Annealing Temp (°C)	Accession Number
Mb102C	(GT)10…(GT)8…(GT)9	Upstream: TGACTGATACTCTGAAGGGTGT Downstream: GCTCCTCTCATTACTGGCTC	53	DQ852336
Mb118H	(GT)23	Upstream: TGTCAAGCACCAACCTCC Downstream: GTGCGTATTGAAGTGATGAGA	54	DQ852335
Mb116H	(GT)23	Upstream: TGCGTATTGAAGTGATGAGA Downstream: GCTGTCAAGCACCAACCT	59–49	DQ852334
Mb43	(GT)22	Upstream: TGGTGGCTGTTACCCTAT Downstream: AAACCTGCATCTCCTGAA	56.9	EF599347
Mb41	(AC)3A(AC)13A(AC)9	Upstream: GGACTATCAGCCCACCTCT Downstream: TTCTTAACCACTGGACCACC	53.3	EF599346
Mb40	(GT)15GC(GT)7…(GT)9GC(GT)5	Upstream: CACCTAGTGGCGATTTCA Downstream: AACAGAGGGCGGTTGGAT	56.9	EF599345
Mb39	(GT)34	Upstream: ATCAAACCCACATCTCCT Downstream: TGCCCTGGTTAGAACTCC	56.9	EF599344
Mb38	(AC)14…(AC)14	Upstream: AGTGAGGCGAGTCTGTGAG Downstream: TCCCGTGTCCAAGAAAGT	60	EF599343
Mb37	(GT)9ATGG(GT)13ATG(GT)9	Upstream: TGTGGGTGAACTCAATCT Downstream: ATGGTATCTGACTCCAATAT	58.2	EF599342
Mb34	(GT)16…(GT)13…CT(GT)6	Upstream: CAACATTTGGGAGGAGGAT Downstream: GTGAGGGCTTCTGGTGAT	57.9	EF599341
Mb33	(GT)26	Upstream: TCCTCGCTGATTATTTGG CGGATTCGTAAAGTGGGT	55.2	EF599340
Mb18	(GT)15	Upstream: CTCCAGGCAAGAACACTG Downstream: GCAAGAAGTTATGCAATCAA	55.2	EF599337

**Table 2 animals-10-00255-t002:** Genetic diversity parameters of the population at 12 microsatellite loci.

Locus	N	A	A_R_	PIC	Ho	H_E_	F_IS_
Mb102C	221	11	6	0.721	0.385	0.746 *	0.486
Mb118H	238	21	16	0.825	0.840	0.843	0.005
Mb116H	238	14	9	0.797	0.811	0.820	0.013
Mb43	234	9	3	0.781	0.603	0.803	0.254
Mb41	94	4	1	0.330	0.383	0.360	−0.065
Mb40	86	16	8	0.880	0.767	0.895	0.143
Mb39	238	11	8	0.477	0.244	0.512 *	0.526
Mb38	238	15	9	0.812	0.824	0.834	0.013
Mb37	237	10	3	0.811	0.878	0.833	−0.052
Mb34	235	9	3	0.796	0.562	0.822 *	0.317
Mb33	238	19	12	0.789	0.660	0.804	0.179
Mb18	237	2	0	0.373	0.827	0.496 *	−0.668
All loci	-	11.75	6.50	0.699	0.649	0.731	0.115

N—Number of individuals; A—Number of alleles ofthe population; A_R_—Number of rare alleles, PIC—Polymorphism richness; H_O_—Observed heterozygosity; H_E_—Expected heterozygosity; F_IS_—Wright’s inbreeding coefficient. *—means significantly deviated from Hardy-Weinberg equilibrium (*p* < 0.01).

**Table 3 animals-10-00255-t003:** Allele diversity of captive forest musk deer populations in 5 successive generations.

Locus	1st Generation					2nd Generation					3rd Generation					4th Generation					5th Generation				
	N	A	A_R_	R_A_	P_R_	N	A	A_R_	R_A_	P_R_	N	A	A_R_	R_A_	P_R_	N	A	A_R_	R_A_	P_R_	N	A	A_R_	R_A_	P_R_
Mb102C	25	7	5.320	4	0	47	8	6.247	2	0	52	7	5.718	3	0	49	9	5.896	4	0	48	10	6.887	4	2
Mb118H	32	15	9.150	9	3	48	16	8.588	11	1	55	12	7.860	5	0	52	14	7.485	10	0	51	15	8.295	11	0
Mb116H	32	10	7.338	6	1	48	10	7.021	3	0	55	10	7.561	3	0	52	10	6.358	5	0	51	12	6.890	8	1
Mb43	31	9	6.982	6	1	48	8	6.359	2	0	54	9	7.383	1	0	52	8	6.704	1	0	50	8	6.500	1	0
Mb41	11	3	2.909	1	0	21	4	3.345	1	0	19	2	1.983	0	0	14	3	2.966	0	0	29	4	3.167	1	0
Mb40	10	8	8.000	0	0	19	13	10.437	3	1	20	12	9.641	3	0	17	11	8.546	5	0	20	12	9.486	3	0
Mb39	32	6	4.011	3	0	48	7	3.933	4	2	55	9	4.590	6	1	52	7	4.312	5	0	51	6	3.789	3	0
Mb38	32	10	6.697	5	0	48	14	8.670	7	1	55	13	6.801	9	1	52	10	6.513	5	0	51	11	7.156	6	0
Mb37	32	8	6.753	3	0	48	9	6.902	2	0	55	8	6.564	1	0	52	10	7.126	5	1	50	9	7.221	2	0
Mb34	32	8	6.356	3	0	46	8	6.493	2	0	54	8	5.571	3	0	52	7	5.953	1	0	51	9	6.700	3	1
Mb33	32	13	8.606	7	1	48	12	7.762	5	0	55	15	8.339	9	1	52	13	7.929	8	0	51	13	7.806	6	1
Mb18	31	2	2.000	0	0	48	2	2.000	0	0	55	2	2.00	0	0	52	2	2.000	0	0	51	2	2.000	0	0
Total	32	99	6.177	47	6	48	111	6.48	42	5	55	107	6.168	43	3	52	104	5.982	49	1	51	111	6.325	48	5

N—Number of individuals; A—Number of alleles; A_R_—Numbers of allelic richness; R_A_—Number of rare alleles; P_R_—Number of private alleles.

**Table 4 animals-10-00255-t004:** Comparison of genetic diversity of 5 successive generations in the closed breeding population.

Locus	1st Generation			2nd Generation			3rd Generation			4th Generation			5th Generation		
	Ho	H_E_	F_IS_	Ho	H_E_	F_IS_	Ho	H_E_	F_IS_	Ho	H_E_	F_IS_	Ho	H_E_	F_IS_
Mb102C	0.6	0.746	0.199	0.362	0.798	0.55	0.385	0.697	0.451	0.449	0.690	0.351	0.229	0.766	0.703
Mb118H	0.844	0.869	0.029	0.854	0.851	−0.004	0.836	0.852	0.018	0.827	0.822	−0.006	0.843	0.83	−0.016
Mb116H	0.938	0.815	−0.153	0.875	0.817	−0.072	0.764	0.849	0.101	0.827	0.808	−0.024	0.706	0.773	0.087
Mb43	0.6	0.769	0.262	0.646	0.780	0.174	0.648	0.848	0.237	0.615	0.798	0.231	0.500	0.801	0.378
Mb41	0.727	0.515	−0.441	0.429	0.373	−0.154	0.263	0.235	−0.125	0.143	0.373	0.626	0.414	0.359	−0.157
Mb40	1.00	0.874	−0.154	0.632	0.919	0.319	0.850	0.906	0.064	0.647	0.848	0.243	0.800	0.900	0.114
Mb39	0.281	0.543	0.486	0.292	0.522	0.444	0.218	0.532	0.592	0.212	0.484	0.565	0.235	0.496	0.528
Mb38	0.813	0.795	−0.022	0.708	0.873	0.19	0.891	0.818	−0.090	0.769	0.816	0.058	0.922	0.843	−0.095
Mb37	1.00	0.839	−0.195	0.833	0.830	−0.005	0.964	0.822	−0.174	0.788	0.836	0.058	0.840	0.838	−0.002
Mb34	0.688	0.823	0.167	0.63	0.842	0.254	0.389	0.765	0.494	0.596	0.809	0.265	0.569	0.850	0.333
Mb33	0.813	0.827	0.018	0.563	0.790	0.290	0.655	0.818	0.201	0.673	0.775	0.132	0.647	0.821	0.214
Mb18	0.613	0.489	−0.258	0.896	0.503	−0.795	0.818	0.502	−0.640	0.846	0.502	−0.697	0.882	0.498	−0.786
All loci	0.743	0.742	0.001	0.643	0.742	0.134	0.640	0.720	0.112	0.616	0.713	0.139	0.632	0.731	0.137

H_O_—Observed heterozygosity, H_E_—Expected heterozygosity, F_IS_—Wright’s inbreeding coefficient.

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
