# Peer review of "Changes in the Population Genetic Structure of Captive Forest Musk Deer (Moschus berezovskii) with the Increasing Number of Generation under Closed Breeding Conditions"

_animals, 2020, doi:10.3390/ani10020255_

Round 1

Reviewer 1 Report

Review of

Changes in the population genetic structure of captive forest musk deer (Moschus berezovskii) with each generation under closed breeding conditions

by

Yonghua Cai, Jiandong Yang, Jianming Wang, Ying Yang, Wenlong Fu, Chengli Zheng, Jianguo Cheng, Yutian Zeng, Yan Zhang, Ling Xu, Yan Ren, Chuanzhi Lu and Ming Zhang

General comments

Generally, the authors raise a very important issue, i.e. the maintenance of high levels of genetic diversity in populations of captive forest musk deer bred for musk production. However, the authors did not transfer the message of why this is important. The introduction needs to provide more background information on musk deer, their value, why they are hunted so intensely and why they are bred in captivity. The authors need to tell a story! In addition, they need to explain that there are many such breeding centres in China and they need to elaborate on genetic diversity and why it is important to keep it high in a captive population. In relation to this they should also very carefully consider the use of domestication! In the introduction the authors need to explain the terms random mating and rotation mating and explain that the latter was already introduced in their study population and that this might be the reason for the high genetic diversity. Given this, the authors need to make predictions at the end of the introduction. Material and methods need to be re-arranged, putting the paragraph 1.3 first. Genetic methods appear to be sound but as an ecologist, I am not really in the position to estimate the correctness of the methods applied. The same applies to the results and the way they were presented. The discussion is merely a comparison of genetic diversity of populations. Hereby it remains often unclear whether the authors refer to other captive populations or to wild populations.

At the end, it remains puzzling to the reader whether the authors have confirmed or rejected their expectations/predictions. As it seems the population studied is neither inbred, not does it seem to have a reduced genetic diversity. Why is it like this and what was that expected?

At places, the English grammar and phrasing needs some revision!

Specific comments

Title

What are closed breeding conditions – you never use this term in the rest of your manuscript. In Table 4 you use the term closed breeding population. Are the two related? What do you intend to describe with this term?

Introduction

P1, last line: I think you should write: … increasing the captive population. Since you have no data and it will be virtually impossible (or at least difficult) to study the genetic diversity of wild populations.

P1: From the late 1960s to the late 1990s, the number of wild musk deer dropped sharply due to illegal hunting and habitat fragmentation from one million animals to about two hundred thousand, which means that it is endangered. Please make a statement on why this sharp decrease in only 30 years. Why are they hunted so much and for what???? Provide reference for musk deer being endangered (prove IUCN conservation category and citation). Rephrase that part of the sentence (which means that it is endangered).

P2, L3: … and to domesticate the timid forestry musk deer: remove or replace the word ‘domesticate’. Domestication is a process that takes place over 100s (1000s) of years and should be definitely not the aim of breeding musk deer – especially not if you consider them to be released into the wild one day.

Maybe better say: and to improve the husbandry conditions of timid forestry musk deer.

P2, L3: social animal: I disagree – they are not really social (rather solitary and territorial), and if you make that claim you must provide a reference.

P2, L4: what population do you mean – do not be vague. What population do you refer to? The overall wild population in China (or in East Asia), or that of all breeding centres in China? Or only that in the breeding centre where this study was carried out? What is the name of te place?

P2L4: easily frightened is the same as timid. Remove this repetition.

P2, third last line: Again, what population do you refer to????

P2: ‘This breeding management system was widely applied in captive musk deer in China’: you need to provide a reference for such a statement.

P3, third last line: … in a species such as the musk deer: musk deer is not a species but a family (Moschidae). Your species is called Forest musk deer (Moschus berezovskii). Please revised your phrasing here!

Also: What are markers of desirable physical traits?: I think you mean something else here – please revise!

P4: ‘The amplified fragment length polymorphism (AFLP) marker technique was used to compare the genetic diversity of two captive musk deer populations in Baisha of Sichuan Province and the Jinfeng Mountains by Zhao et al. [16], Peng et al . and Feng et al. analysed genetic diversity of captive forest musk deer populations in Sichuan and Shaanxi provinces, China, by comparing their mitochondrial DNA loop region’s sequence structure: This is one sentence – please revise and make it two!!

P4: Is the Jinfeng Mountains population investigated by Zhao et al. wild or captive????

P4: Is the captive forest musk deer populations in Miyaluo, Jinfeng Mountain and Barkam the same as the Jinfeng Mountains population investigated by Zhao et al.????

P4: It remains unclear to me why this paragraph was added to the manuscript. What is the benefit of mentioning all these heterozygosity values?

P5: ‘selective breeding of forest musk deer to increase genetic diversity’: why is it important to increase genetic diversity? The authors need to provide more background information on why it is important to keep genetic diversity high. If, as the authors claimed earlier, the animals should be domesticated, a high genetic diversity is not imperative. So why keep genetic diversity high and why avoid inbreeding???

Material and methods

The authors must provide a paragraph termed Study area and species. Here they need to describe the breeding centre, the purpose of the captive population and what animals are kept there (species, number, sex ratio, age distribution). When did wild caught individuals last supplement the population? Do they exchange individuals with other breeding centres? Explain the terms random mating and rotation mating and explain that the latter was already introduced in this captive population but in others not.

P3(1.3): ‘Their offspring , 3 year old male and 2.5 year old female musk deer began to breed, and the number of male musk deer which were selected to mate is determined by the number of female musk deer in offspring population (according to female : male = 15:2 ), and the ranking of high musk yield’:

This sentence is grammatically not correct and does not make sense – what do the authors what to say? Please revise.

P3: Better say: Other male musk deer in offspring population were not used for breeding but only for musk production.

P3: Thise offspring population must to avoid sib pair mating and parent offspring mating, and also obey the BMS RM rules. Spelling and grammar need to be revised.

P3 (1.3) ‘Breeding management system’ should be the first paragraph under Material and Methods!

Results

P2.2: Please revise the entire paragraph (difficult to understand): e.g. With the introduction of individuals to participate in the …: what individuals were introduced? It should be: breeding management system.

What is the message of this sentence: At the same time, the number of rare alleles (less than 5% of the population) will be the highest as 49 were found in 4th generation group. 49 what??? Alleles??

P3: What are the last two young generations????

Discussion

P3.1(P1): The authors must be concise and consistent: why are the musk deer populations endangered, why are the hunted so much and why are they kept in such large number in captivity? For conservation? I don’t think so –mainly because of musk production. What is musk? Why is it so precious? This needs to be outlined in the Introduction.

Why is it important to have a high genetic diversity? Only within your breeding centre or in all Chinese breeding centres? If you want to domestic the captive musk deer (as stated earlier in the introduction) you do not need to bother about genetic diversity but rather about targeted breeding towards a higher musk yield. The authors need to make this distinction very carefully (in the Introduction but also here).

P1: Dujiangyan, Miyaluo: are these two different places or do these names refer to the same breeding centre?

P1: In order to save the endangered musk deer: Again, musk deer is a genus or family, but I think you are referring to a certain species (Moschus berezovskii) and you should write it like this. Be precise in your wording!!!

L26/27: Better say: In our study, twelve microsatellite loci were used to detect polymorphisms in a captive forest musk deer population, and the amplified result ….

L29: Xia et al.[2] studied genetic diversity in a population of 48 forest musk deer: Is that a captive or a wild population. The authors must be more careful with their wording. It is important to know whether the population investigated is wild or captive. Otherwise the statement provides NO information to the reader.

L31: Dujiangyan of Sichuan Province: Again – wild or captive?

L33: In case readers from other countries than China read your paper they do not know Miyaluo, the Jinfeng Mountains and Barkamou. You need to make very clear what is what!! Miyaluo is a breeding centre located in the Jinfeng Mountains????

L37: ‘Other deer populations have lower diversity…’: captive or wild??????

L40: Do you want to say: The high genetic diversity of forest musk deer unrevealed in this study was also supported by a different marker known as ….????

L40-52: Why provide all these details but with no relation to your own findings.

L60: ‘ … is at risk of shifting the genetic diversity from six to two years of…: what do you mean – please revise and make it clear. How you shift genetic diversity from one generation to another one??

L85: Better say: … increasing the genetic diversity of captive forest musk deer populations, ….

L85: … and keeping the loss of habitat to a minimum …: habitat loss is not really the reason for the decline of musk deer species in the wild. Please explain the reader already in the introduction what the problem is (i.e. hunting for musk which is used in Chinese traditional medicine)

L87-90: ‘Judging from the inbreeding coefficient increment (ΔFIS), the breeding management system decreased the risk of inbreeding compared to the mating system of random mating within the breeding group, and the non-rotation mating of male musk deer in different generations, but it also increased the degree of inbreeding compared to random mating in a large population’: Mind your grammar – better make it two sentences!

Last paragraph: please clarify/explain the use of terms: random mating, rotation mating. Random mating and rotation mating are used in the same breeding group? That sounds contradictory – please clarify! What is missed mating of females?? Why not propose the introduction of stud book like in other endangered species kept in captivity. This stud book should include individuals from all breeding centres in the country.

In the long term it will be required to exchange males between breeding centres in order to maintain the high level of genetic diversity.

Author Response

We thank the reviewer’s comment very much, and in the introduction , we added the importance of maintaining the genetic diversity of musk deer and the reasons for the decrease of the population of wild musk deer. As well as the current status of breeding management system and breeding center of forest musk deer in China.

We put “1.3 Breeding Management System” to the beginning of the “materials and methods” section, and we also provide more background information of the forest musk deer and describe in more detail the breeding management system

In the discussion, when we compare the genetic diversity, we add the explanation of whether the population is captive or wild musk deer.

After revision, the manuscript has been edited at Pivot Sciedit (Certificate No.: pvsc-19100101-3) by an editor who is a native English-speaker and who has an English language qualification.

For the questions raised by the reviewer, we will answer one-to-one as follows:

Specific comments

Title

Point 1: What are closed breeding conditions – you never use this term in the rest of your manuscript. In Table 4 you use the term closed breeding population. Are the two related? What do you intend to describe with this term?

Response 1: Yes“closed breeding conditions”is related to “closed breeding population”. “The closed breeding condition” means that there are no male and female musk deer to be induced into the population”. We added this explanation in 1.1 section.

Introduction

Point 2: P1, last line: I think you should write: … increasing the captive population. Since you have no data and it will be virtually impossible (or at least difficult) to study the genetic diversity of wild populations.

Response 2:Yes,it should be the captive population, we have revised it. To express more accurately in "captive forest musk deer", "captive population", "wild forest musk deer" and "wild population", we carefully checked the whole text and further increased the scientificity of the expression.

Point 3: P1: From the late 1960s to the late 1990s, the number of wild musk deer dropped sharply due to illegal hunting and habitat fragmentation from one million animals to about two hundred thousand, which means that it is endangered. Please make a statement on why this sharp decrease in only 30 years. Why are they hunted so much and for what???? Provide reference for musk deer being endangered (prove IUCN conservation category and citation). Rephrase that part of the sentence (which means that it is endangered).

Response 3: We have increased the reasons for the sharp decrease in the number of forest musk deer, and also increased the description of the reasons for the endangered wild forest musk deer.

Musk produced by male musk deer has high value, and it is used as a precious natural flavor and an important ingredient of traditional Chinese medicines. Therefore, wild male forest musk deer were illegally hunted for collecting musk and habitat fragmentation. From the late 1960s to the late 1990s, the number of wild musk deer dropped sharply from one million animals to about two hundred thousand

We provide reference for musk deer being endangered

Harris, R. 2016. Moschus chrysogaster . The IUCN Red List of Threatened Species 2016: e.T13895A61977139. http://dx.doi.org/10.2305/IUCN.UK.2016-1.RLTS.T13895A61977139.en.

Point 4: P2, L3: … and to domesticate the timid forestry musk deer: remove or replace the word ‘domesticate’. Domestication is a process that takes place over 100s (1000s) of years and should be definitely not the aim of breeding musk deer – especially not if you consider them to be released into the wild one day.

Maybe better say: and to improve the husbandry conditions of timid forestry musk deer.

Response 4:  we change “domesticate” into “improve the husbandry conditions of timid forestry musk deer”.

Point 5: P2, L3: social animal: I disagree – they are not really social (rather solitary and territorial), and if you make that claim you must provide a reference.

Response 5: We have changed“social animal”into“territorial animal”.

Point 6:P2, L4: what population do you mean – do not be vague. What population do you refer to? The overall wild population in China (or in East Asia), or that of all breeding centres in China? Or only that in the breeding centre where this study was carried out? What is the name of the place?

Response 6: We have modified the expression. Here we just show that in a large captive population, it’s difficult to record genealogy of offspring if male and female musk deer mate randomly. It’s not a specific population.

Point 7: P2L4: easily frightened is the same as timid. Remove this repetition.

Response 7: We have deleted the repetition.

Point 8:P2, third last line: Again, what population do you refer to????

Response 8:It is captive population. We describe the breeding management system of captive forest musk deer.

We have revised it. To express more accurately in "captive forest musk deer", "captive population", "wild forest musk deer" and "wild population", we carefully checked the whole text and further increased the scientificity of the expression.

Point 9: P2: ‘This breeding management system was widely applied in captive musk deer in China’: you need to provide a reference for such a statement.

Response 9:We have added references, which are in Chinese.

Point 10:P3, third last line: … in a species such as the musk deer: musk deer is not a species but a family (Moschidae). Your species is called Forest musk deer (Moschus berezovskii). Please revised your phrasing here!

Response 10:We have changed“the musk deer”into“Forest musk deer”.

Point 11:Also: What are markers of desirable physical traits?: I think you mean something else here – please revise!

Response 11:We have changed“desirable physical traits”into“valuable traits”.

Point 12:P4: ‘The amplified fragment length polymorphism (AFLP) marker technique was used to compare the genetic diversity of two captive musk deer populations in Baisha of Sichuan Province and the Jinfeng Mountains by Zhao et al. [16], Peng et al . and Feng et al. analysed genetic diversity of captive forest musk deer populations in Sichuan and Shaanxi provinces, China, by comparing their mitochondrial DNA loop region’s sequence structure: This is one sentence – please revise and make it two!!

Response 12: We added the conjunction "and" and modified it into two paratactic sentences.

The amplified fragment length polymorphism (AFLP) marker technique was used to compare the genetic diversity of two captive musk deer populations in Baisha of Sichuan Province and the Jinfeng Mountains by Zhao et al [16], and Peng et al. [17,18] and Feng et al. [19] analyzed genetic diversity of captive forest musk deer populations in Sichuan and Shaanxi provinces, China, by comparing their mitochondrial DNA-loop region’s sequence structure.

Point 13:P4: Is the Jinfeng Mountains population investigated by Zhao et al. wild or captive????

Response 13:Jinfeng Mountains population is captive. Baisha and Jinfeng Mountains are two farms of captive forest musk deer in Dujianyan, Sichuan Province, China

The amplified fragment length polymorphism (AFLP) marker technique was used to compare the genetic diversity of two captive musk deer populations in Baisha of Sichuan Province and the Jinfeng Mountains by Zhao et al [16].

Point 14:P4: Is the captive forest musk deer populations in Miyaluo, Jinfeng Mountain and Barkam the same as the Jinfeng Mountains population investigated by Zhao et al.????

Response 14:They were the same as population in Zhao’ study, but the genetic structure analysis of population uses different techniques. First of all, AFLP marker technique was used to compare the genetic diversity of two captive musk deer populations in Baisha of Sichuan Province and the Jinfeng Mountains by Zhao et al. then Zhao used SSR technique to analyze Baisha and Jinfeng

Please see the reference 16.

[16]Zhao, S.; Chen, X.; Wan, Q. Assessment of genetic diversity in the forest musk deer (Moschus berezovskii) using microsatellite and AFLP markers. Chin. Sci. Bull. 2011, 56: 2565-2569.

Point15:P4: It remains unclear to me why this paragraph was added to the manuscript. What is the benefit of mentioning all these heterozygosity values?

Response 15: The heterozygosity and the number of alleles in the population reflect genetic diversity and genetic variation of population to a certain extent. We show Zhao's results to prove the high genetic diversity of the population and to compare them with the results we will get.

  If the reviewer thinks it is unnecessary, we delete it.

Point 16:P5: ‘selective breeding of forest musk deer to increase genetic diversity’: why is it important to increase genetic diversity? The authors need to provide more background information on why it is important to keep genetic diversity high. If, as the authors claimed earlier, the animals should be domesticated, a high genetic diversity is not imperative. So why keep genetic diversity high and why avoid inbreeding???

Response 16:We have the background information about genetic diversity, and we discussed the importance of genetic diversity from three aspects: species protection, inbreeding control and breeding to improve musk yield.

  However, captive populations are faced with the problem of inbreeding, loss of genetic diversity and decreased behavioral fitness. Also, high genetic diversity of captive forest musk deer is the basis of artificial breeding for increasing musk yield.

Material and methods

Point 17:The authors must provide a paragraph termed Study area and species. Here they need to describe the breeding centre, the purpose of the captive population and what animals are kept there (species, number, sex ratio, age distribution). When did wild caught individuals last supplement the population? Do they exchange individuals with other breeding centres? Explain the terms random mating and rotation mating and explain that the latter was already introduced in this captive population but in others not.

Response 17:We have added the information of breeding centre and the captive population. We chose the blood sample from the closed breeding population, therefore no captured individuals are added to this population, and no individual was exchanged with other breeding center.

  The forest musk deer were housed and raised in the farm of captive forest musk in Songgang town, Barkam, Sichuan, and there are more than 300 forest musk deer in Barkam center for Genetic protection and musk production. The generations of these individuals were recorded, and their blood samples were collected at the time of musk collection or at the time of vaccination. These blood samples have been collected continuously for nearly 20 years.

  Random mating means that 17 forestry musk deer (Male: 2; Female: 15) within a breeding group were housed in the same shed, and the male and female musk mate freely.

  Rotation mating means that the 2 male forest musk deer were rotated to another shed for mating with other 15 female musk deer in the next breeding season.

  “Random mating” and “Rotation mating” is the term in Breeding management system of rotated mating (BMS-RM). We re-described the BMS-RM.

Musk deer are seasonal breeding animals, and the breeding season lasts from October until the next February. During the breeding season, 17 forestry musk deer (Male: 2; Female: 15) within a breeding group were housed in the same shed, and the male and female musk mate freely. The 2 male forest musk deer were rotated to another shed for mating 15 female forest musk deer in the next breeding season. In the next generation, offspring of 3 year old male and 2.5 year old female musk deer begin to be used reproduction, and the number of male musk deer which are selected to mate is determined by the number of female musk deer in the offspring population (according to female : male = 15:2). The offspring male individuals with high musk yield are preferentially selected for mating, and Other male musk deer in offspring population were not used for breeding but only for musk production.. All offspring female individuals are used for reproduction. The Breeding management system of rotated mating (BMS-RM) can avoid the mating of the same male and female musk deer within at least five years. The breeding of all musk deer in Barkam center is strictly obeyed the BMS-RM rules, and all individuals avoid sib-pair mating and parent-offspring mating.

Point 18:P3(1.3): ‘Their offspring , 3 year old male and 2.5 year old female musk deer began to breed, and the number of male musk deer which were selected to mate is determined by the number of female musk deer in offspring population (according to female : male = 15:2 ), and the ranking of high musk yield’:

This sentence is grammatically not correct and does not make sense – what do the authors what to say? Please revise.

Response 18:We revised the sentence.

In the next generation, their offspring of 3 year old male and 2.5 year old female musk deer began to be used reproduction, and the number of male musk deer which were selected to mate is determined by the number of female musk deer in the offspring population. The offspring male individuals with high musk yield are preferentially selected for mating, and other male individuals are used only for musk production. All female are used for reproduction.

We want to clarify the breeding management system (BMS) and generation of forest musk deer.

Point 19:P3: Better say: Other male musk deer in offspring population were not used for breeding but only for musk production.

Response 19:We have revised it according to the reviewer's suggestion.

Point 20:P3: This offspring population must to avoid sib pair mating and parent offspring mating, and also obey the BMS-RM rules. Spelling and grammar need to be revised.

Response 20:We have revised the sentence.

The breeding of all musk deer in Barkam center is strictly obey the BMS-RM rules, and all individuals avoid sib-pair mating and parent-offspring mating.

Point 21:P3 (1.3) ‘Breeding management system’ should be the first paragraph under Material and Methods!

Response 21:We have put the “Breeding management system” to the first paragraph in Material and Methods

Results

Point 22:P2.2: Please revise the entire paragraph (difficult to understand): e.g. With the introduction of individuals to participate in the …: what individuals were introduced? It should be: breeding management system.

Response 22:The explanation for the increase and loss of private alleles is wrong in the old manuscript, and we revised it.

    There will be a largest number of private alleles (PR = 6) present in the 1st generation compared to other subsequent generations, and the number of private alleles continuously decreased from the 1st to 4th generation because some musk offspring didn't participate in the breeding in BMS-RM, the number of private alleles increased from the 4st to 5th generation because rotation mating induced new male musk deer from other shed. The number of private alleles might vary up and down around a constant value, if more generations are examined because the breeding obey BMS-RM within the closed large population. The number of rare alleles (less than 5% of the population) will be the highest were found in 4th generation group (RA = 49). However, the risk of loss of private and rare alleles is always present, as some males are not involved in breeding.

Point 23:What is the message of this sentence: At the same time, the number of rare allele genes (less than 5% of the population) will be the highest as 49 were found in 4th generation group. 49 what??? Alleles??

Response 23:It means that there are 49 rare alleles in 12 MMS loci. We optimized the expression. We have revised it.

The number of rare alleles (less than 5% of the population) will be the highest were found in 4th generation group (RA = 49).

Point 24:P3: What are the last two young generations????

Response 24:We change “the last two young generations” into “the 4th and 5th generation

  The minimum at 1st generation was 0.001, and the degree of inbreeding increased from the 2nd generation, especially in the 4th (FIS = 0.137) and 5th (FIS = 0.139) generation

Discussion

Point 25:P3.1(P1): The authors must be concise and consistent: why are the musk deer populations endangered, why are the hunted so much and why are they kept in such large number in captivity? For conservation? I don’t think so –mainly because of musk production. What is musk? Why is it so precious? This needs to be outlined in the Introduction.

Response 25:We strongly support the reviewer's suggestion. This part should not be put in the discussion, but in the introduction. We have added reference and greatly modified the introduction.

  The forest musk deer (Moschus berezovskii) is an Asian ungulate that is listed as an endangered species by the International Union for Conservation of Nature (IUCN). It is included in the CITES Appendix II and is a Level 2 protected wild animal in China. It is widely distributed in many provinces in China and in the northern part of Vietnam. Musk produced by male musk deer has high value, and it is used as a precious natural flavor and an important ingredient of traditional Chinese medicines. Therefore, wild male forest musk deer were illegally hunted for collecting musk and habitat fragmentation. From the late 1960s to the late 1990s, the number of wild musk deer dropped sharply from one million animals to about two hundred thousand.

The breeding of captive forest musk deer originated in 1958 in China. The early breeding technique was introduced mainly to expand the population, explore the musk collection method and to improve the husbandry conditions of timid forestry musk deer. After the non-invasive technique for musk collection was successfully developed and the Chinese government commenced captive breeding program for musk harvesting in 1958, some breeding centers of captive forest musk deer, such as Barkam, Miyaluo, Dujianyan, Fengxian, Yaan, and other places have been established for protecting wild forest musk deer and providing essential musk for traditional Chinese medicine. However, captive populations are faced with the problem of inbreeding, loss of genetic diversity and decreased behavioral fitness. Also, high genetic diversity of captive forest musk deer is the basis of artificial breeding for increasing musk yield. Among these centers, Barkam center was the first to be established, consequently, studying the genetic diversity of captive forest musk deer in Barkam center is of great importance for maintaining genetic diversity, expanding the population, and improving the musk yield of male population.

Point 26:Why is it important to have a high genetic diversity? Only within your breeding centre or in all Chinese breeding centres? If you want to domestic the captive musk deer (as stated earlier in the introduction) you do not need to bother about genetic diversity but rather about targeted breeding towards a higher musk yield. The authors need to make this distinction very carefully (in the Introduction but also here).

Response 26:We have re-described the importance of high genetic diversity in introduction, and the discussion of importance of high genetic diversity in discussion had been deleted.

Point 27:P1: Dujiangyan, Miyaluo: are these two different places or do these names refer to the same breeding centre?

Response 27: they are different place and also are different population of captive forest musk deer.

Point 28:P1: In order to save the endangered musk deer: Again, musk deer is a genus or family, but I think you are referring to a certain species (Moschus berezovskii) and you should write it like this. Be precise in your wording!!!

Response 28: Thank the reviewer. We have deleted the first paragraph of the discussion, and re-described the scientific significance of the study.

Point 29:L26/27: Better say: In our study, twelve microsatellite loci were used to detect polymorphisms in a captive forest musk deer population, and the amplified result ….

Response 29: We have revised it according to the reviewer's suggestion, and added “captive”.

Point 30:L29: Xia et al.[2] studied genetic diversity in a population of 48 forest musk deer: Is that a captive or a wild population. The authors must be more careful with their wording. It is important to know whether the population investigated is wild or captive. Otherwise the statement provides NO information to the reader.

Response 30:This is captive population.

To express more accurately in "captive forest musk deer", "captive population", "wild forest musk deer" and "wild population", we carefully checked the whole text and further increased the scientificity of the expression.

Point 31:L31: Dujiangyan of Sichuan Province: Again – wild or captive?

Response 31:This is captive population.

To express more accurately in "captive forest musk deer", "captive population", "wild forest musk deer" and "wild population", we carefully checked the whole text and further increased the scientificity of the expression.

Point 32:L33: In case readers from other countries than China read your paper they do not know Miyaluo, the Jinfeng Mountains and Barkamou. You need to make very clear what is what!! Miyaluo is a breeding centre located in the Jinfeng Mountains????

Response 32:We strongly support the reviewer's suggestion. We have added the information of Miyaluo, the Jinfeng Mountains and Barkam in the introduction. They are different center of captive forest musk deer.

Point 33:L37: ‘Other deer populations have lower diversity…’: captive or wild??????

Response 33:This is wild population.

To express more accurately in "captive forest musk deer", "captive population", "wild forest musk deer" and "wild population", we carefully checked the whole text and further increased the scientificity of the expression.

Point 34:L40: Do you want to say: The high genetic diversity of forest musk deer unrevealed in this study was also supported by a different marker known as ….????

Response 34:

We deleted the comparison of the method of genetic diversity detection.

Judged from PIC, these captive forest musk deer have high genetic diversity, and this study also confirmed captive forest musk deer have high genetic diversity.

Point 35:L40-52: Why provide all these details but with no relation to your own findings.

Response 35:These previous results show that the genetic diversity of captive forest musk deer in China has maintained a high level, which reflects the effectiveness of inbreeding control in the artificial breeding management system of captive forest musk deer in China. And our results showed the genetic diversity of captive forest musk deer in China has maintained a high level, and it proves that the BMS-RM is effective in inbreeding controlling.

  Therefore, I think these previous studies are related to ours. We have added the discussion.

Point 36:L60: ‘ … is at risk of shifting the genetic diversity from six to two years of…: what do you mean – please revise and make it clear. How you shift genetic diversity from one generation to another one??

Response 36:We have clarified the change, it really is slowly down.

Captive musk deer population in Barkam center is at risk of genetic diversity decrease from the 1st generation to the 5th generation and increasing the degree of inbreeding.

Point 37:L85: Better say: … increasing the genetic diversity of captive forest musk deer populations, ….

Response 37:We have revised it according to the reviewer's suggestion, and added “captive”.

Point 38:L85: … and keeping the loss of habitat to a minimum …: habitat loss is not really the reason for the decline of musk deer species in the wild. Please explain the reader already in the introduction what the problem is (i.e. hunting for musk which is used in Chinese traditional medicine)

Response 38: We have explained the reason for the decline of musk deer species in the wild in introduction, the main reason is hunting for musk which is used in Chinese traditional medicine.

We also revised the discussion.

  The captive breeding program is our main hope for protecting the existing gene pool, increasing the genetic diversity of musk deer populations, being forbidden to hunt wild musk deer, and breeding more captive forest musk deer to provide enough musk for traditional Chinese medicine.

Point 39:L87-90: ‘Judging from the inbreeding coefficient increment (ΔFIS), the breeding management system decreased the risk of inbreeding compared to the mating system of random mating within the breeding group, and the non-rotation mating of male musk deer in different generations, but it also increased the degree of inbreeding compared to random mating in a large population’: Mind your grammar – better make it two sentences!

Response 39:We optimized the sentence to make its structure clearer.

Judging from the inbreeding coefficient increment (ΔFIS), our breeding management system (BMS-RM) decreased the risk of inbreeding compared with the mating system of random mating within the breeding group, and with the non-rotation mating system of male musk deer across different generations, but it also increased the degree of inbreeding compared with the random mating system in a large population.

Point 40:Last paragraph: please clarify/explain the use of terms: random mating, rotation mating. Random mating and rotation mating are used in the same breeding group? That sounds contradictory – please clarify! What is missed mating of females?? Why not propose the introduction of stud book like in other endangered species kept in captivity. This stud book should include individuals from all breeding centres in the country.

Response 40:

“Random mating” means 17 forest musk deer (2 male + 15 female = a breeding group) freely mate in breeding season.

“Rotation mating” means the 2 male forest musk deer were rotated to another breeding group for mating in the next breeding season.

Accurate stud book is impossible for captive forest musk deer. The mother of the offspring can record clearly, but the father of the offspring is not clear. Two male musk deer in the breeding group may be the father of the offspring.

Artificial insemination of musk deer has not been successful, and nature mate is the only way to breeding. Forest musk deer is easy to be frightened, so it is impossible to artificially control the mating between a male musk deer and an estrous female musk deer. The male and female musk deer must live together for more than 3 months and familiarize them with each other before the breeding season. If a male and a female are kept together for a long time, the number of male musk is not enough to breeding. If only one male musk deer in the breeding group, the conception rate of female musk deer is very low. Therefore, it is necessary to put two or more male musk deer in a breeding group. If there are too many male musk deer in the breeding group, the father of the offspring is not clear, and male musk deer fight each other for mating and cause damage. Therefore, in the breeding process of forest musk deer, two male musk deer and 15 female musk deer are raised in a shed and form a breeding group. The 2 male musk are changed into another shed in the next breeding season.

We show a schematic diagram for understanding “random mating” and “Rotation mating”

Point 41:In the long term it will be required to exchange males between breeding centres in order to maintain the high level of genetic diversity.

Response 41:We strongly agree with the reviewers, and exchange male between breeding center is an effective way to improve genetic diversity. However, our objective is to assess whether genetic diversity of captive musk deer in Barkam center under closed breeding condition by obeying BMS-RM rules. In our study, exchange male is not the means we consider.

Reviewer 2 Report

The study of Cai et al. deals with an important topic in wildlife conservation: the need of protecting the genetic diversity of captive population of endangered species by improving breeding management systems to avoid inbreeding. In general, the study is catchy and the rationale is there, however I think that the manuscript, as it is now, needs some significant improvements on the explanation of data analysis, presentation of the results and the quality of discussion. Also, the flow of the Introduction seems to have, now and then, some missing links giving the impression of an incomplete picture. Whereby, major revisions are still necessary to make the paper acceptable for publication.

Specific comments are listed below:

P1-6: Lines are missing from page 1 to page 6. According to the Journal guidelines, lines must be provided on the manuscript. Indeed, this is a useful tool to facilitate the review process. Lines only appear at page 7 of the manuscript when the authors present Tables 3 and 4.

Abstract:

P1, L 4 of the abstract: Please, consider to change ‘utilization’ with ‘usage’. P1, L 4-5 of the abstract: Please, consider to modify the sentence ‘the results of screening the captive forest musk deer..’’ with ‘the results of the screening of the captive forest musk deer..’. P1: I would suggest to include P-values rather then just descriptive data to make the abstract more straightforward for the readers.

Introduction:

P2, L 6-8: I think the link between a drop in population and the importance of studying genetic diversity in captive forestry musk deer is missing. There is a jump from info about the status of the wild population to the study of genetic diversity in captive populations. I suggest to provide info that helps the reader to go from one topic to the other. The authors briefly introduce these notions in their Discussion section (L10-16), whereby something similar should also be said in the Introduction to set the scene and to justify the reason of their study. The study was carried out on a captive population, why? Is there any conservation programs in progress? If so, why? What type of issues captive populations have to face with? What is the role of genetic diversity for ex situ conservation programs? Etc. In addition, following to the new Intro, I would also suggest to move the sentence ‘Consequently,…increasing population’ (L 7-8) after the sentence ‘..for animal breeding programs’ (L26-27) to explain the importance of studying genetic diversitya and its role. P2, L 6 of the Introduction: I guess including a phrase such as ‘which means that it is endangered’ is kind of superfluous and repetitive. Please, do consider to remove it. P2, L 13-18: The breeding management system briefly described in the text needs to be referenced in order to understand why and who established this type of management for captive forestry musk deer in China. P2, L 29-33: Please, consider to rephrase this sentence as too long and repetitive. P2, L 40: Please, consider to add ‘a’ before ‘high genetic diversity’. P3, L 44-49: I think a clear statement about the aim of the study and the authors’ hypothesis is missing.

Materials and Methods:

P3, L 1-2 of M&M: A statement about the Ethical approval for the use of animals in research is missing. Please, the authors should check the ethical guidelines of the journal (‘Manuscripts containing original descriptions of research conducted in experimental animals must contain details of approval by a properly constituted research ethics committee. As a minimum, the project identification code, date of approval and name of the ethics committee or institutional review board should be cited in the Methods section’) and include the missing info. P3, Paragraph 1.1: When is the study carried out? P3, Paragraph 1.1: This sentence is way too long. Please, consider to rephrase it as suggested: ‘The 238 forest musk deer blood samples used in this study were collected from the captive population of musk deer at the musk deer farm of Sichuan Institute of Musk Deer Breeding in Barkam of Sichuan Province. All individuals were from 5 successive generations (1st generation: 32; 2nd generation: 48; 3rd generation: 55; 4th generation: 52, and 5th generation: 51) and were housed in 21 separated breeding groups (2 males and 15 females musk deer in each breeding group).’ P3, L8: Please, consider to add ‘using’ between ‘by’ and ‘the phenol-chloroform..’. 15.P3, Paragraph 1.3: I suggest to make this paragraph the first of the section ‘Materials and Methods’. Indeed, providing firstly information on the breeding management system applied to the studied population, and secondly info on data collection, methodologies and data analysis would make the flow of the story more logical for the readers. Please, if possible, do also provide references on the breeding management system. P4, L 34: Please, modify ‘Thise’ with ‘This’. P4, Paragraph 1.4: Please, include more specific information on the type of statistical analysis performed (e.g. a non-parametric analysis ‘Wilcoxon test’) and why did you decide to carry out that analysis? For instance, were data checked for normality, etc.? I found a bit confusing that some notions on statistical analysis were only mentioned in the Results section instead of M&M (e.g. L 12 and 35 of Results). P3 and 4: Overall, I would suggest a general English editing throughout the text.

Results:

P5, L 4-5 of Results: I am wondering why did the authors decide to mention locus Mb41 as the one with the lowest number of individuals genotyped (which is 94 not 93 according to Table 2 of the manuscript), when there is locus Mb40 which has 86 individuals identified genotypically? P5, L 20: Please, add ‘s’ at the word ‘generation’ of the paragraph 2.2. P6, L 33-35: The authors state that the heterozygosity of the 5th generation group was the lowest among generations. However, according to Table 4 of the manuscript, the 4th generation is the one holding the lowest heterozygosity (both observed – 0.616 vs 0.632 - and expected – 0.713 vs 0.731). Please, do provide an explanation or consider to modify the text. P6, L 34: I suppose the numerical data were switched. In fact, 0.742 is the HE while 0.743 is the HO of the 1st generation group according to Table 4. P6, L 36: What about the P-values of HO and FIS? Was there any significant difference between generations? P6, L 40-41: This sentence is more discussion material. P6, L 43: I would suggest to include an extra table to provide data of the 21 breeding groups as for those given between generations (i.e. Tables 3 and 4). This may help to make the overall results more comprehensible to the readers. P6, L 42-44: Also, I would consider to better present the expected vs. actual FIS for population, breeding groups and breeding management systems by including the info in one sentence. Indeed, at the moment data are spread all-over the results section (e.g. the actual FIS of the population is in L9-10 while the expected FIS is in L 42).

Discussion:

P 9, L 13-16: I think that these notions should also be included in the Introduction section as stated above in my previous comment (See comment 5). P9, L 25: I am a bit confused about the context of the words ‘with the improved enriched library’ in the sentence L 24-26. Please, consider to rephrase it. P9, L27-37: It seems to me that the authors are citing too many examples to discuss one single concept. P9, L 37-39: Are these studies on other deer populations related to captive or wild species? My suggestion to the authors is to discuss the possible motivations behind these differences among deer species (e.g. habitat, management systems, conservation programs etc.) instead of only presenting data saying that red deer, sika deer etc. have lower diversity than musk deer. P9, L40-41: Is the AFLP a marker or a technique? Please, be precise with the terminology. P9-10, L 40-52: The authors discussed about the AFLP studies and their outcomes as presented by the literature. However, no results on AFLP data related to their study were provided by the authors. Did the authors use the AFLP tecnique in their study? If so, please rephrase the text in M&M and Results because this is not fully clear to me. P10, L 64: I am a bit confused because I cannot find the data ‘8.25’ for the 1st generation and ‘9.25’ for the 5th generation related to the number of alleles. Is Table 3 the one to look at for finding these results?. P10, L 69-75: Data discussed in these lines sound a bit contradictory to me. How is possible that the authors firstly say that Huang et al. found similar conclusions to their study (i.e. as the age decreased, the number of private alleles also decreased), while few lines after (L74-75) they say that the 5th generation group of their study had more private alleles? Please, consider to explain this info. P10, L79: Please, modify this reference (Ballou, 2005) as for the Journal guidelines. A number has to be given instead of citing surname and year of the paper. P11, L 368: Please, consider to rephrase the sentence as I found quite tough to understand what the authors are arguing about.

Conclusions:

P11, L 102-103: This section is missing. According to the journal guidelines (See ‘Research Manusctip Sections on the Journal website) this section is mandatory. Please, include it in the manuscript. P11, L 105-106: Part of the back matter (i.e. Author contributions and Conflicts of interest) is missing. Please, include this info by following the Journal guidelines.

Author Response

We thank the reviewer’s comment very much, and we have improved introduction, material and method, results and discussion according to the reviewer’s commending.

After revision, the manuscript has been edited at Pivot Sciedit Service (Certificate No.: pvsc-19100101-3) by an editor who is a native English-speaker and who has an English language qualification.

For the questions raised by the reviewer, we will answer one-to-one as follows:

Specific comments are listed below:

Point 1:P1-6: Lines are missing from page 1 to page 6. According to the Journal guidelines, lines must be provided on the manuscript. Indeed, this is a useful tool to facilitate the review process. Lines only appear at page 7 of the manuscript when the authors present Tables 3 and 4.

Response 1:I’m sorry for this, now we have added the line number.

Abstract:

Point 2:P1, L 4 of the abstract: Please, consider to change ‘utilization’ with ‘usage’.

Response 2:We have revised it.

The data provide a new molecular basis for the evaluation and usage of the breeding management system.

Point 3: P1, L 4-5 of the abstract: Please, consider to modify the sentence ‘the results of screening the captive forest musk deer..’’ with ‘the results of the screening of the captive forest musk deer..’.

Response 3: We have revised it.

The results of screening of the captive forest musk deer population revealed significant genetic diversity.

Point 4:P1: I would suggest to include P-values rather then just descriptive data to make the abstract more straightforward for the readers.

Response 4: This is good suggestion, and we add the P-value

 In the experiment, H-W-E was analyzed by Chi-square test, and F-statistical analysis on the data showed that the genetic diversity of population. The result shows that genetic diversity of population decrease, but the difference with generation-increasing isn’t significant, but the inbreeding coefficient significant increased, therefore, we added “FIS in the 1st generation is significant lower in the 5th generation (P < 0.05)” in results.

   Also, we added the F-statistical in 1.4 data analysis.

Introduction:

Point 5:P2, L 6-8: I think the link between a drop in population and the importance of studying genetic diversity in captive forestry musk deer is missing. There is a jump from info about the status of the wild population to the study of genetic diversity in captive populations. I suggest to provide info that helps the reader to go from one topic to the other.

Response 5: Another reviewer put up the similar question. We have increased literature on the reasons for the decline of musk deer population and the importance of maintaining the genetic diversity of captive forest musk deer.

   The forest musk deer (Moschus berezovskii) is an Asian ungulate that is listed as an endangered species by the International Union for Conservation of Nature (IUCN)[1]. It is included in the CITES Appendix II and is a Level 2 protected wild animal in China [2]. It is widely distributed in many provinces in China and in the northern part of Vietnam [3]. Musk produced by male musk deer has high value, and it is used as a precious natural flavor and an important ingredient of traditional Chinese medicines. Therefore, wild male forest musk deer were illegally hunted for collecting musk and habitat fragmentation. From the late 1960s to the late 1990s, the number of wild musk deer dropped sharply from one million animals to about two hundred thousand [4,5].

The breeding of captive forest musk deer originated in 1958 in China. The early breeding technique was introduced mainly to expand the population, explore the musk collection method and to improve the husbandry conditions of timid forestry musk deer [4]. After the non-invasive technique for musk collection was successfully developed and the Chinese government commenced captive breeding program for musk harvesting in 1958 [6], some breeding centers of captive forest musk deer, such as Barkam, Miyaluo, Dujianyan, Fengxian, Yaan, and other places have been established for protecting wild forest musk deer and providing essential musk for traditional Chinese medicine [7]. However, captive populations are faced with the problem of inbreeding, loss of genetic diversity and decreased behavioral fitness. Also, high genetic diversity of captive forest musk deer is the basis of artificial breeding for increasing musk yield [8]. Among these centers, Barkam center was the first to be established, consequently, studying the genetic diversity of captive forest musk deer in Barkam center is of great importance for maintaining genetic diversity, expanding the population, and improving the musk yield of male population.

Point 6:The authors briefly introduce these notions in their Discussion section (L10-16), whereby something similar should also be said in the Introduction to set the scene and to justify the reason of their study. The study was carried out on a captive population, why? Is there any conservation programs in progress? If so, why? What type of issues captive populations have to face with? What is the role of genetic diversity for ex situ conservation programs? Etc.

Response 6: I think the first paragraph of the discussion is put in the introduction to explain the importance of maintaining the genetic diversity of captive forest musk deer. So we've put the first paragraph of our discussion in the introduction. And we also added new reference to provide the background information for the study.

Point 7: In addition, following to the new Intro, I would also suggest to move the sentence ‘Consequently,…increasing population’ (L 7-8) after the sentence ‘..for animal breeding programs’ (L26-27) to explain the importance of studying genetic diversitya and its role.

Response 5: We revised it,and the introduction has a big change. Please see the Response 5.

Point 8: P1, L 6 of the Introduction: I guess including a phrase such as ‘which means that it is endangered’ is kind of superfluous and repetitive. Please, do consider to remove it.

Response 5: We have deleted it.

  The forest musk deer (Moschus berezovskii) is an Asian ungulate that is listed as an endangered species by the International Union for Conservation of Nature (IUCN)[1]. It is included in the CITES Appendix II and is a Level 2 protected wild animal in China [2]. It is widely distributed in many provinces in China and in the northern part of Vietnam [3]. Musk produced by male musk deer has high value, and it is used as a precious natural flavor and an important ingredient of traditional Chinese medicines. Therefore, wild male forest musk deer were illegally hunted for collecting musk and habitat fragmentation. From the late 1960s to the late 1990s, the number of wild musk deer dropped sharply from one million animals to about two hundred thousand [4,5].

Point 9:P2, L 13-18: The breeding management system briefly described in the text needs to be referenced in order to understand why and who established this type of management for captive forestry musk deer in China.

Response 9: we added the reference, and re-described the BMS-RM in 1.1 Breeding Mangement System in materials and methods. BMS-RM rule was established by the Sichuan Institute of Musk Deer Breeding, and was widely used at the farm of captive musk deer in China.

Point 10:P2, L 29-33: Please, consider to rephrase this sentence as too long and repetitive.

Response 10: we have changed two sentences.

  The amplified fragment length polymorphism (AFLP) marker technique was used to compare the genetic diversity of two captive musk deer populations in Baisha of Sichuan Province and the Jinfeng Mountains by Zhao et al. [16]. Peng et al. [17,18] and Feng et al. [19] analyzed genetic diversity of captive forest musk deer populations in Sichuan and Shaanxi provinces, China, by comparing their mitochondrial DNA-loop region’s sequence structure.

Point 11:P2, L 40: Please, consider to add ‘a’ before ‘high genetic diversity’.

Response 11: We have added ‘a’ before ‘high genetic diversity.

Point 12:P3, L 44-49: I think a clear statement about the aim of the study and the authors’ hypothesis is missing.

Response 12: We have strengthened our statement of purpose.

  In the study, the objective was to evaluated genetic structure of the population at the musk deer farm of the Sichuan Institute of Musk Deer Breeding in Barkam and examined the change in genetic structure of successive generations based on polymorphisms at twelve microsatellite loci, and, we also compared the distribution of private alleles and inbreeding coefficient in the population in different generations and provided a scientific reference for assessing the breeding management system.

Materials and Methods:

Point 13:P3, L 1-2 of M&M: A statement about the Ethical approval for the use of animals in research is missing. Please, the authors should check the ethical guidelines of the journal (‘Manuscripts containing original descriptions of research conducted in experimental animals must contain details of approval by a properly constituted research ethics committee. As a minimum, the project identification code, date of approval and name of the ethics committee or institutional review board should be cited in the Methods section’) and include the missing info.

Response 13: We have added the information

All operations strictly obey Wildlife Protection Acts and Regulations of China, and also were approved by the Wildlife Protection Committee of Musk Deer Research Institute (MDRI-2009-02), and the Ethics Committee of Use of Endangered Wild Animals in Research and Teaching (2018YSZH0019).

Point 14:P3, Paragraph 1.1: When is the study carried out?

Response 14: These blood samples have been collected continuously for nearly 20 years and were kept at -80 °C, and the study was performed in 2018.

We have described in 1.1 Animals and Breeding management system.

Point 15:P3, Paragraph 1.1: This sentence is way too long. Please, consider to rephrase it as suggested: ‘The 238 forest musk deer blood samples used in this study were collected from the captive population of musk deer at the musk deer farm of Sichuan Institute of Musk Deer Breeding in Barkam of Sichuan Province. All individuals were from 5 successive generations (1st generation: 32; 2nd generation: 48; 3rd generation: 55; 4th generation: 52, and 5th generation: 51) and were housed in 21 separated breeding groups (2 males and 15 females musk deer in each breeding group).’

Response 15: we revised it according to the reviewer’s suggestion.

We chose 238 forest musk deer blood samples with generations record and used in this study, and these sample were collected from the captive population of musk deer in Barkam of Sichuan Province. All individuals (Female: 142, Male: 96) were from 5 successive generations (1st generation: 32; 2nd generation: 48; 3rd generation: 55; 4th generation: 52, and 5th generation: 51).

Point 16:P3, L8: Please, consider to add ‘using’ between ‘by’ and ‘the phenol-chloroform..’. 15.

Response16: we have changed “by” into “using”.

The blood samples were preserved at Ë—80°C after addition of citric acid-EDTA as anticoagulant. Genomic DNA was extracted using the phenol-chloroform method [24] and stored at Ë—20°C.

Point 17:P3, Paragraph 1.3: I suggest to make this paragraph the first of the section ‘Materials and Methods’. Indeed, providing firstly information on the breeding management system applied to the studied population, and secondly info on data collection, methodologies and data analysis would make the flow of the story more logical for the readers. Please, if possible, do also provide references on the breeding management system.

Response 17: Another reviewer put up the similar question. We have revised it. In the revised manuscript, “1.1Animals and Breeding management system” had been as the first paragraph of materials and methods. We have added the reference for Breeding Management System.

1.1    Animals and Breeding management system

The forest musk deer were housed and raised in the farm of captive forest musk in Songgang town, Barkam, Sichuan, and there are more than 300 forest musk deer in Barkam center for Genetic protection and musk production. The generations of these individuals were recorded, and their blood samples were collected at the time of musk collection or at the time of vaccination. These blood samples have been collected continuously for nearly 20 years, and were kept at Ë—80°C. All operations strictly obey Wildlife Protection Acts and Regulations of China, and also were approved by the Wildlife Protection Committee of Musk Deer Research Institute (MDRI-2009-02), and the Ethics Committee of Use of Endangered Wild Animals in Research and Teaching (2018YSZH0019).

Musk deer are seasonal breeding animals, and the breeding season lasts from October until the next February. During the breeding season, 17 forestry musk deer (Male: 2; Female: 15) within a breeding group were housed in the same shed, and the male and female musk mate freely. The 2 male forest musk deer were rotated to another shed for mating 15 female forest musk deer in the next breeding season. In the next generation, offspring of 3 year old male and 2.5 year old female musk deer begin to be used reproduction, and the number of male musk deer which are selected to mate is determined by the number of female musk deer in the offspring population (according to female : male = 15:2). The offspring male individuals with high musk yield are preferentially selected for mating, and other male musk deer in offspring population were not used for breeding but only for musk production. All offspring female individuals are used for reproduction. The Breeding management system of rotated mating (BMS-RM) can avoid the mating of the same male and female musk deer within at least five years. The breeding of all musk deer in Barkam center is strictly obeyed the rules of BMS-RM, and all individuals avoid sib-pair mating and parent-offspring mating.

To help you better understand the breeding management system, we give you the following diagram:

Point 18:P4, L 34: Please, modify ‘Thise’ with ‘This’.

Response 18: we have corrected this mistake.

Point 19:P4, Paragraph 1.4: Please, include more specific information on the type of statistical analysis performed (e.g. a non-parametric analysis ‘Wilcoxon test’) and why did you decide to carry out that analysis? For instance, were data checked for normality, etc.? I found a bit confusing that some notions on statistical analysis were only mentioned in the Results section instead of M&M (e.g. L 12 and 35 of Results).

Response 19: We added “Wilcoxon test” and “F-test” into the 1.4 data analysis.

FIS of all generation in at 12 microsatellite loci were compared by F-test. HE of different generation was compared by the Wilcoxon nonparametric test using GENEPOP 3.4 [26].

Point 20:P3 and 4: Overall, I would suggest a general English editing throughout the text.

Response 20: After revision, the manuscript has been edited at Pivot Sciedit Service (Certificate No.: pvsc-19100101-3) by an editor who is a native English-speaker and who has an English language qualification.

Results:

Point 21:P5, L 4-5 of Results: I am wondering why did the authors decide to mention locus Mb41 as the one with the lowest number of individuals genotyped (which is 94 not 93 according to Table 2 of the manuscript), when there is locus Mb40 which has 86 individuals identified genotypically?

Response 21: The description of this result is not rigorous or wrong, we have revised it.

238 individuals were genotyped in the Mb118H, Mb116H, Mb39, Mb38 and Mb33 loci, but only 86 individuals were identified genotypically in the Mb40 locus, and 94 individuals were identified genotypically in the Mb41.

Point 22: P5, L 20: Please, add ‘s’ at the word ‘generation’ of the paragraph 2.2.

Response 22: we have revised it.

Point 23:P6, L 33-35: The authors state that the heterozygosity of the 5th generation group was the lowest among generations. However, according to Table 4 of the manuscript, the 4th generation is the one holding the lowest heterozygosity (both observed – 0.616 vs 0.632 - and expected – 0.713 vs 0.731). Please, do provide an explanation or consider to modify the text.

Response 23: This is a wrong description, we have modified the text. Maybe, the comparison of heterozygosity between 1st generation and 5th generation revealed the change of genetic diversity, which caused the wrong description.

Point 24:P6, L 34: I suppose the numerical data were switched. In fact, 0.742 is the HE while 0.743 is the HO of the 1st generation group according to Table 4.

Response 24: we carefully checked the number and recalculated the number of all loci in table 4. These figures are right.

This is a wrong description, we have modified the text, and we have deleted it.

Point 25:P6, L 36: What about the P-values of HO and FIS? Was there any significant difference between generations?

Response 25: we have added the P-value of HO and FIS.

    The Wilcoxon nonparametric test showed that there was no significant difference in HE between the 1st and 5th generation (P>0.05).

The minimum at 1st generation was 0.001, and the degree of inbreeding significantly increased compared with the 2nd (FIS = 0.134), the 3rd (FIS = 0.112) to the 4th (FIS = 0.137) and 5th (FIS = 0.139) generation ( P < 0.01)

Point 26:P6, L 40-41: This sentence is more discussion material.

Response 27: Actually, these discussions on the genetic diversity of captive forest musk deer from various places give reader misunderstanding if there is no summarization to these discussions. We have added the objective of these discussions.

    The results suggested the genetic diversity of captive forest musk deer maintained a high level compared with other endangered wild cervidae animals (red deer, sika deer and white-tailed deer).

    These results of previous studies show that the genetic diversity of captive forest musk deer in China has maintained a high level, which reflects the effectiveness of inbreeding control in the artificial breeding management system of captive forest musk deer in China. And our results showed the genetic diversity of captive forest musk deer from the closed population in Barkam center maintained a high level, and it proved that the BMS-RM was effective in inbreeding controlling.

Point 27: P6, L 43: I would suggest to include an extra table to provide data of the 21 breeding groups as for those given between generations (i.e. Tables 3 and 4). This may help to make the overall results more comprehensible to the readers.

Response 27:

  In old manuscript, the description of animals is inappropriate in materials and methods. In the revised manuscript, we re-described animals.

 We chose 238 forest musk deer blood samples with generation record.These blood samples have been collected continuously for nearly 20 years, and were kept at Ë—80°C.

Therefore, 238 individuals were not from 21 breeding group, but from 5 successive generation.

Point 28:P6, L 42-44: Also, I would consider to better present the expected vs. actual FIS for population, breeding groups and breeding management systems by including the info in one sentence. Indeed, at the moment data are spread all-over the results section (e.g. the actual FIS increment of the population is in L9-10 while the expected FIS is in L 42).

Response 28:

The reviewer raised a good question, putting similar results together. FIS (the population inbreeding coefficient) in L9-10 was gotten by microsatellite analysis, and ΔFIS (FIS increment) was counted by FIS. We adjusted the result.

FIS= the population inbreeding coefficient (in L9-10)

ΔFISE=expected FIS increment (in L42-44)

ΔFISA= actual FIS increment (in L42-44).

The population inbreeding coefficient (FIS) showed a gradual increase from the 1st generation to the 5th generation. The minimum at 1st generation was 0.001, and the degree of inbreeding significantly increased compared with the 2nd (FIS = 0.134), the 3rd (FIS = 0.112)to the 4th (FIS = 0.137) and 5th (FIS = 0.139) generation ( P < 0.01). The expected FIS increment (ΔFISE1) of the random mating large population was 0.005, and the expected FIS increment (ΔFISE2) of the breeding group was 0.050, while actual FIS increment (ΔFISA) in the breeding management system was 0.034. In order to maintain diversity in a captive population, the inbreeding of close relatives must be prevented.

Discussion:

Point 28:P 9, L 13-16: I think that these notions should also be included in the Introduction section as stated above in my previous comment (See comment 5).

Response 28: We revised it, and please see point 6 and response 6.

Point 29:P9, L 25: I am a bit confused about the context of the words ‘with the improved enriched library’ in the sentence L 24-26. Please, consider to rephrase it.

Response 29: it means an improved and high quality genome library of musk deer was used in Zhao’s study for screening the highly polymorphic microsatellite loci, and then these highly polymorphic microsatellite loci were used to study the genetic diversity of captive or wild. We revised it and make it easier to be understood.

   Zhou et al [13] identified eight highly polymorphic microsatellite loci that could be used with the improved genomic library of forest musk deer to study the genetic diversity of captive or wild (Moschus berezovskii).

Point 30:P9, L27-37: It seems to me that the authors are citing too many examples to discuss one single concept.

Response 30: In the old manuscript, this discussion does not indicate whether the musk deer is wild or captive. Therefore the reviewer thought it unnecessary.

  In the revised manuscript, we noted that these forest musk deer are captive.

In the introduction, we added these farms of captive forest musk deer in China, and in discussion, we discussion the genetic diversity of captive forest musk deer in Miyaluo, Dujiangyan, Jinfeng Montains, and Barkam. Also we discussed the genetic diversity of wild cervidae animals (red deer, sika deer and white-tailed deer).

The previous results suggested the genetic diversity of captive forest musk deer maintained a high level compared with other endangered wild cervidae animals (red deer, sika deer and white-tailed deer), which reflects the effectiveness of inbreeding control in the artificial breeding management system of captive forest musk deer in China.

The objective of these discussions have been summarized in the revised manuscript.

Point 31:P9, L 37-39: Are these studies on other deer populations related to captive or wild species? My suggestion to the authors is to discuss the possible motivations behind these differences among deer species (e.g. habitat, management systems, conservation programs etc.) instead of only presenting data saying that red deer, sika deer etc. have lower diversity than musk deer.

Response 31: All studies in discussion, forest musk deer are captive. We also added the possible motivations of low genetic diversity of other deer species.

The results suggested the genetic diversity of captive forest musk deer maintained a high level compared with other endangered wild cervidae animals (red deer, sika deer and white-tailed deer). The impossible reasons is that the population of captive forest musk deer is larger, and to take measures of inbreeding control, while the number of wild deer is small, habitat fragmentation and free mating lead to a higher degree of inbreeding.

Point 32: P9, L40-41: Is the AFLP a marker or a technique? Please, be precise with the terminology.

Response 32: It is technique for molecular marker. We have revised it

    The amplified fragment length polymorphism (AFLP) was first applied to measure genetic diversity of two captive musk deer populations, one in Baisha of Sichuan and the other at a reserve in the Jinfeng Mountains [16].

Point 33:P9-10, L 40-52: The authors discussed about the AFLP studies and their outcomes as presented by the literature. However, no results on AFLP data related to their study were provided by the authors. Did the authors use the AFLP tecnique in their study? If so, please rephrase the text in M&M and Results because this is not fully clear to me.

Response 33: because the referenc

We detect genetic diversity using microsatellite in the study. We have deleted the data (908 AFLP marker to detect 22 pairs of selective primers), and only kept the result: both populations had high genetic diversity.

The amplified fragment length polymorphism (AFLP) was first applied to measure genetic diversity of two captive musk deer populations, one in Baisha of Sichuan and the other at a reserve in the Jinfeng Mountains, and both populations had high genetic diversity [16,22], but the Jinfeng Mountain group was highest [16].

Point 34:P10, L 64: I am a bit confused because I cannot find the data ‘8.25’ for the 1st generation and ‘9.25’ for the 5th generation related to the number of alleles. Is Table 3 the one to look at for finding these results?.

Response 34: The value is the average number of alleles in 12 MS loci. They didn’t show in Table 3, and was counted.

Point 35:P10, L 69-75: Data discussed in these lines sound a bit contradictory to me. How is possible that the authors firstly say that Huang et al. found similar conclusions to their study (i.e. as the age decreased, the number of private alleles also decreased), while few lines after (L74-75) they say that the 5th generation group of their study had more private alleles? Please, consider to explain this info.

Response 35: In the old manuscript, Age is wrong, and generation is right. In the revised manuscript, we revised it.

Huang et al.[23] came to a similar conclusion in their study of young vs adult captive groups in Miyaluo center. With generation-increasing, the number of private alleles decreased. Therefore, it is necessary that the breeding management level of the captive population is gradually improving.

Point 36: P10, L79: Please, modify this reference (Ballou, 2005) as for the Journal guidelines. A number has to be given instead of citing surname and year of the paper.

Response 36: we deleted “Ballou, 2005” and give the number [27].

Point 37:P11, L 368: Please, consider to rephrase the sentence as I found quite tough to understand what the authors are arguing about.

Response 37: We have revised the sentence, and made it easier to be understood.

In order to decrease the risk of inbreeding and ΔFIS, the appropriate number of individuals within each breeding group, the number of breeding groups from the large population, and the rotation mating of male musk deer from each breeding group in different generations were considered as important factors. A large population with the certain number of individuals (N), the number of individuals within each breeding group (NBG), and the number of breeding groups from the large population () that are necessary to be further clarified in order to achieve the minimum inbreeding increment using the breeding management system under closed breeding condition.

Conclusions:

Point 38:P11, L 102-103: This section is missing. According to the journal guidelines (See ‘Research Manusctip Sections on the Journal website) this section is mandatory. Please, include it in the manuscript.

Response 38: We added the Conclusions in the revised manuscript.

Point 39:P11, L 105-106: Part of the back matter (i.e. Author contributions and Conflicts of interest) is missing. Please, include this info by following the Journal guidelines.

Response 39: we added these parts, such as, Author Contributions, Funding, and Conflicts of Interest according to the Journal guidelines.

Reviewer 3 Report

In this manuscript authors examined the changes in the population genetic structure of captive forest musk deer (Moschus berezovskii) with generation-increasing to assess the Breeding Management System. The study provides an important reference for the breeding system of musk deer to control inbreeding, and It provides scientific significance for breeding and management strategies of wild animals. In the discussion, the author puts forward a significant problem: if a large population is divided into many subgroups, free-mating within subgroups, and rotated-mating of males among subgroups, how to control inbreeding?

Author Response

We thank the reviewer very much for his comment and affirmation of our study.

Point1:In this manuscript authors examined the changes in the population genetic structure of captive forest musk deer (Moschus berezovskii) with generation-increasing to assess the Breeding Management System. The study provides an important reference for the breeding system of musk deer to control inbreeding, and It provides scientific significance for breeding and management strategies of wild animals. In the discussion, the author puts forward a significant problem: if a large population is divided into many subgroups, free-mating within subgroups, and rotated-mating of males among subgroups, how to control inbreeding?

Response 1:We response to the question “how to control inbreeding?”

We obey rules of Breeding Management System of Rotated Mating (BMS-RM) in the process of breeding to achieve control inbreeding.

The rule of BMS-RM is below:

Musk deer are seasonal breeding animals, and the breeding season lasts from October until the next February. During the breeding season, 17 forestry musk deer (Male: 2; Female: 15) within a breeding group were housed in the same shed, and the male and female musk mate freely. The 2 male forest musk deer were rotated to another shed for mating 15 female forest musk deer in the next breeding season. In the next generation, offspring of 3 year old male and 2.5 year old female musk deer begin to be used reproduction, and the number of male musk deer which are selected to mate is determined by the number of female musk deer in the offspring population (according to female : male = 15:2). The offspring male individuals with high musk yield are preferentially selected for mating, and Other male musk deer in offspring population were not used for breeding but only for musk production.. All offspring female individuals are used for reproduction. The Breeding management system of rotated mating (BMS-RM) can avoid the mating of the same male and female musk deer within at least five years. The breeding of all musk deer in Barkam center is strictly obeyed the BMS-RM rules, and all individuals avoid sib-pair mating and parent-offspring mating.

Round 2

Reviewer 1 Report

see attached file

Author Response

Dear reviewer:

We thank your comments very much, and response all question and suggestion in General comments and Specific comments by one-to one as follows:

General comments

The authors have improved the manuscript by adhering to the comments and suggestions of the two reviewers (e.g. omit the term domestication from the manuscript). However, the authors need to be more careful and neat with regard phrasing, grammar and sentence construction!! The authors claimed to have consulted Pivot Sciedit (Certificate No.: pvsc-19100101-3) and an editor who is a native English-speaker and who has an English language qualification. I strongly recommend to consult Pivot Sciedit again or to change to another editor! Moreover, the authors need to be more careful and neat with regard to punctuations, spelling and sentence construction.

Response 1: The manuscript has been edited by an editor who is a native English-speaker in England after the manuscript was revised according to the reviewer’s comment. Because language-editor is not experts in genetics, we can't guarantee the accuracy of the use of professional terms.

At the end of the introduction the authors need to make a prediction(s). What did the authors expect with regard to genetic diversity? What other studies were carried out previously (e.g. Zhao et al. [21], Peng et al [22,23] and Feng et al [24]), and what were their finding with regard to genetic diversity? What are your predictions based on their findings? A potential predictions could be: In case diversity is high in your population the current breeding management is sufficient, or in case diversity is low breeding rotation needs to be extended to other breeding centres within China. The authors must make it clear why it is important to carry out his study!!!

Response 2: It’s a good question, and we added the potential prediction in the introduction.

    If there is a high genetic diversity in the closed population, which imply the present BMS is effective in maintain the stability of genetic structure. If the inbreeding coefficient does not increase with increasing number of generation, which imply the present BMS is effective in avoiding inbreeding. Otherwise, the BMS needs to be improved in the future.

Genetic methods appear to be sound but as an ecologist, I am not really in the position to estimate the correctness of the methods applied. The same applies to the results and the way they were presented. The discussion was improved but still needs to give a clear recommendation with regard to future captive breeding of this species (see below: stud book, exchange of males between Chinese breeding centres).

Response 3: This is a good suggestion, and the exchange of males between Chinese breeding centres is effective measure to improve genetic diversity. We added the recommendation in discussion.

Specific comments

Abstract

Point 1: L30: What is Barkam? Please provide a full name here, e.g. Barkam Musk Deer Breeding Centre at the Sichuan Forest Research Institute.

Response 1: Yes, Barkam is place name, herein; it’s Barkam Musk Deer Breeding Centre at the Sichuan Forest Research Institute.

We have revised it.

Introduction

Point 2: L49-50: Therefore, wild male forest musk deer were illegally hunted for collecting musk and habitat fragmentation. This sentences makes no sense and is grammatically wrong: male FMD cannot be hunted for habitat fragmentation! Please revise?

Response 2: We have revised it.

Therefore, wild male forest musk deer were illegally hunted for collecting musk.

Point 3: L57-58: … have been established for protecting wild forest musk deer…. Revise: musk deer breeding centres were established to relieve wild populations from the severe hunting pressure and provide markets with legal musk obtained from captive animals. Only if the authors explain this context the reader can understand why breeding centres help to protect wild populations!!

Response 3: We have revised it.

Some breeding centers of musk deer, such as Barkam, Miyaluo, Dujianyan, Fengxian, Yaan, have been established to relieve wild populations from the severe hunting pressure and provide markets with legal musk obtained from captive animals for traditional Chinese medicine [7].

Point 5: L64: and improving the musk yield of male population. This is a repetition (see previous sentence). Please revise this newly written paragraph and improve grammar and wording!!

Response 5: We have revised it.

Consequently, studies on the genetic diversity of captive forest musk deer in Barkam center is of great importance for maintaining genetic diversity, expanding the population, and assessing the Breeding Management System of the captive forest musk deer.

Point 6: L66: Better say: Forest musk deer is a solitary, territorial species that is easily frightened. Provide a REFERENCE!

Response 6: We revised it according to the reviewer’s suggestion, and added a new reference.

Point 7: L68-73: What you describe here is the opposite of a closed system. Do you refer to musk deer within the same breeding centre as one population? So rotation is taking place within each centre but not between breeding centres? This is what you call a closed system – make it clearer to the unexperienced reader.

Response 7: A closed breeding system, it mean the population or the breeding center  didn’t induced forest musk deer form other centers.

Yes, in the study, these musk deer within the same breeding centre are as one population, and rotation only is performed within each centre but not between breeding centres. We added the description about the closed breeding system.

    “In the next breeding season, the male musk deer were rotated to the other breeding group within the population, and the population didn’t induce forest musk deer form other breeding centers as a closed breeding population.”

Point 8: L74: … a closed large population. What is a closed population? You have not explained that term.

Response 8: Please see response 7, we added the description about the closed breeding system/condition.

Point 9: Why large? Large is relative – omit that attribute.

Response 9: Yes, large is relative. In the study, the number of individuals in large population is much more than the number of individuals in breeding group (17 individuals = 2 male + 15 female).

We deleted “large”, and then check the whole of manuscript. We changed “large population” into “population” in where we think it should be changed.

Point 10: L79: The species is called forest musk deer, not forestry musk deer!!!

Response 10: we have revised it.

Point 11: L87: Shaanxi Provinces

Response 11: we have revised it

Point 12: L89: Guan et al. [27]: a dot after et al.

Response 12: we have added the “.” after “et al”, and check the whole manuscript.

Point 13: L99: It should be: In this study, the/our objective …

Response 13: we have revised it.

In this study, our objective was to evaluated genetic structure of the population at the musk deer farm of Sichuan Institute of Musk Deer Breeding in Barkam and examined the change in genetic structure of successive generations based on polymorphisms at twelve microsatellite loci. And then we also compared the distribution of private alleles and inbreeding coefficients of different generations and provided a scientific reference for the inbreeding control of breeding management system in the population.

Point 14: L101: sentence incomplete

Response 14: We revised the sentence, and please see the response 13.

Point 15: L102: and inbreeding coefficient: make it plural – coefficients

Response 15: we have revised it

Point 16: L102-103: WHY did you also compare the distribution of private alleles and inbreeding coefficient in the population in different generations and provided a scientific reference for assessing the breeding management system?

Response 16: We compare the distribution of private alleles and inbreeding coefficients of different generations, and then assess the

Point 17: L104ff: Here you should make a prediction. What did the authors expect with regard to genetic diversity? What other studies were carried out previously and what was their finding with regard to genetic diversity? In your previous response letter you state: The heterozygosity and the number of alleles in the population reflect genetic diversity and genetic variation of population to a certain extent. We show Zhao's results to prove the high genetic diversity of the population and to compare them with the results we will get. Mention this in your introduction and include this idea into your prediction. Zhao’s study was a snap-shot, but you want to investigate whether genetic diversity changes over generations.

Response 17: We have added the prediction.

Yes, we investigate the current status of genetic diversity of captive forest musk deer, and the changes with generations-increasing in our closed breeding condition. We have revised the objective of the studies.

We expect the genetic diversity is high (PIC > 0.5), and inbreeding coefficients don’t significantly increased with generation-increasing. If the result is our expectation, which implies the BMS-RM can maintain the genetic diversity of the population under the closed breeding population.

Please see Response 3 in General comments

Material & methods

Point 18: L108: What is genetic protection and why do you write it in capital? Please revise. Please write Sichuan Province.

Response 18: “genetic protection” should is “the diversity of genetic resource”. We have revised it.

Point 19: L111-114: This is an ethical statement and should be presented separately at the end of the method section or at the end of the manuscript (in accordance with the journals guidelines).

Response 19: We have revised it, and we also present the ethical statement in the manuscript according to the journals guidelines.

Point 20: L117: … and the male and female musk mate freely. What is a musk? I think you mean musk deer?

Response 20: yes, we have revised it.

“The male and female forest musk deer mate freely.”

Point 21: L118: Better say: The two male forest musk deer were transferred to another shed for mating with another 15 females in the next breeding season.

Response 21: we have revised it.

Point 22: L125: The Breeding management system ….: why capital???

Response 22: We considered it as a proper noun, or used the abbreviation “BMS” in the next text. We have changed “Breeding management system” into “Breeding Management System”.

Point 23: L107-128: You need to revise your English grammar in this newly written paragraph!!

Response 23: We have revised the paragraph.

Point 24: L129: Better say: We obtained blood samples from 238 forest musk deer with a known pedigree … Omit the rest of the sentence (repetition).

Response 24: We think thatgeneration record” is different with “pedigree”, and we have deleted the repetition.

    In the BMS-RM, the father of the offspring is not clear.

Point 25: L154 (183): You need to mention Wright’s inbreeding coefficient at this point. Later you mention it in the result section without having introduced the term in the method section.

Response 25: We have revised it.

Wright’s inbreeding coefficient (FIS), and population differentiation coefficient (FST) of each microsatellite locus were calculated with the FSTAT 2.9.3 software package.

Results

Point 26: L161: random mating large population: poor English and grammatically incorrect

Response 26: We have revised it.

“Large population of random mating”.

Point 27: L176: Within the whole captive population, only four microsatellite loci (Mb102C, Mb39, Mb34, Mb18) out of the twelve deviated from Hardy Weinberg equilibrium (P< 0.01), while the remaining eight were in balance. Please revise the grammar of this sentence!! The way it stands is incorrect.

Response 27: We have revised it.

    Four microsatellite loci (Mb102C, Mb39, Mb34, Mb18) significantly deviated from Hardy-Weinberg equilibrium (P< 0.01), and the remaining eight were in balance.

Point 28: L192: What is a musk offspring? Please be precise and call it musk deer (also at other places within the manuscript).

Response 28: We have revised it, and checked the same mistake in the whole manuscript.

Point 29: L195: …. if more generations are examined because the breeding obey BMS RM within the closed large population. . Omit one of the two full stops. What is a closed large population (see my comment to L74).

Response 29: We have revised it.

A closed breeding system, it mean the population or the breeding center didn’t induced forest musk deer form other centers.

Point 30: L196: At the same time, the number of 196 rare alleles (less than 5% of the population) will be the highest were found in 4th generation group 197 (RA = 49). Revise grammar and wording (i.e. use of two tenses in one sentence: ‘will be’ versus ‘were’).

Response 30: We have revised it.

Point 31: L210ff: The result suggested Breeding Management System of 210 Rotated Mating (BMS-RM) is effective in inbreeding control. In order to maintain diversity in a 211 captive population, the inbreeding of close relatives must be prevented. For the practical application of your data it would be good for how many generations this high genetic diversity can be maintained? There will be a certain point in the future when the BMS-RM is not sufficient any longer and the population needs to be supplement with fresh blood from another breeding centre or from the wild. This should be mentioned in the discussion section!

Response 31: This is a good suggestion, and we added the discussion.

 In our study, 5 successive generations were examined, and we confirmed the high genetic diversity can be maintained within 5 successive generations. We think that how many generations can maintain high genetic diversity is a significant but uncertain problem. Our study can only give a conclusion under the current conditions. The result of study implies the BMS-RM is effective in inbreeding control and in maintaining high genetic diversity.

Point 32: Table 2 (L153): Be careful with the use of the word population. Until L153 you referred to population as all musk deer living in the Barkam Breeding Centre and in L69 you divided this population into many breeding groups. Now, you refer to population as one of the separated breeding units. Please be consistent and do not assign the term population to breeding groups (e.g. L183 A = number of alleles for each population (locus??); L153: the genetic diversity parameters of each population).

Also in Table 3: Allele diversity of captive forest musk deer populations in 5 successive generations. You have investigated only one population, but distinguished between different breeding groups/units. In the left column you show loci, but not populations!!

Response 32: We have revised it, and check the whole manuscript. Surely, “Breeding group” and “population” is different in the study. We confirmed the rightness of “Breeding group” and “population” in the manuscript.

The genetic diversity parameters of the population, including numbers of alleles (A), observed heterozygosity (Ho), expected heterozygosity (HE), and polymorphism information content (PIC) were determined using CEVUS 3.0 software[25]

A = number of alleles of the population

Point 33: In Table 4 your wording is correct: …. in a closed breeding population.

Response 33: we have revised it.

 “in the closed breeding population”.

Discussion

Point 34: L236: Botstein et al[34]: insert a dot and an empty space after et al.!

Response 34: We have revised it, and checked the whole manuscript.

Point 35: L240: Better say: of captive or wild Moschus berezovskii, OR: of captive or wild forest musk deer.

Response 35: We have revised it according to the reviewer’s suggestion.

    Zhou et al. identified eight highly polymorphic microsatellite loci that could be used with the improved genomic library of captive or wild forest musk deer to study the genetic diversity [18].

Point 36: L244: Xia 244 et al[3]: as above

Response 36: we have inserted a dot after “et al”.

Point 37: L250: A = 12.91, HE = 0.899, and PIC = 0.884. Do these values relate to all three breeding centres.

Response 37: Yes, these values are relate to three breeding centres

Point 38: L251: Here you call it sub-populations: be consistent with the terms population, sub-population and breeding unit/group throughout the manuscript!!!

Response 38: We checked the reference, and the author use subpopulation. Judged from the materials and method in the reference, the subpopulation is a sampling group from different centers.

Point 39: L254: Other wild deer ….: Why other wild populations – your is not wild and all musk deer

Response 39: Herein, other wild deer is from references [35], [36] and [37].

Point 40: populations you refer to are captive! Moreover, you need to provide more information about these alleged wild deer populations: did the undergo a bottle neck, how large is the area they live in, etc?

Response 40: We added some information about the population in the study. The population originated from 18 wild forest musk deer in 1958, and some wild individuals were added to the population until 1980. The number of captive forest musk deer in the population is increased from 1980, but there is very little information about wild populations.

Point 41: L258: red deer and white-tailed deer are not endangered, sika deer also only certain populations! Revise this sentence and provide references.

Response 41: We deleted “endangered”, and the references are [35], [36] and [37].

“The results suggested the genetic diversity of captive forest musk deer maintained a high level compared with other wild cervidae animals (red deer, sika deer and white-tailed deer).”

    This is author's discussion, so there is no direct list of references.

Point 42: L259: Impossible???? I think you mean the possible reason is that …..?!

Response 42: Yes, it should be “possible”.

Point 43: L259: To what population do you refer here? Again, the use of the term population is inconsistent. What population do you refer to? The total world population of captive forest musk deer? Larger than what? Larger than the wild red deer population? This is definitely wrong!! Please omit or revise this sentence.

Response 43: we revised the discussion.

The possible reasons is to take measures of inbreeding control in the population of captive forest musk deer, while the population of wild deer live in fragmentation habitat to lead to a high degree of inbreeding.

In this manuscript, we define these individuals from curtain center as a population. And we have reconfirmed “group” “population” in the whole manuscript.

Point 44: L259-61: The impossible reasons is that the population of captive forest musk deer is larger, and to take measures of inbreeding control, while the number of wild deer is small, habitat fragmentation and free mating lead to a higher degree of inbreeding. This sentence is grammatically totally wrong – please revise and make clear what you want to say. Maybe make it two or three sentences.

Response 44: we have revised it.

   The possible reasons is to take measures of inbreeding control in the population of captive forest musk deer, while the population of wild deer live in fragmentation habitat to lead to a high degree of inbreeding.

Point 45: L262-65: Please revise this sentence as well – it is too long and confusing.

Response 45: We have improved the sentence.

The AFLP was first applied to measure genetic diversity of two populations of captive musk deer populations in Baisha of Sichuan Province and  at a reserve in the Jinfeng Mountains, and both populations had high genetic diversity [21,27], but the population in Jinfeng Mountain was higher than that in Baisha [21].

Point 46: L267: Feng et al [19] – please use correct spelling (et al. {19]). Moreover [19] is Fuji, K.; Kobayashi, K.; Hasegawa, O.; Coimbra, M.R.M.; Sakamoto, T.; Okamoto, N. Identification of a single major genetic locus controlling the resistance to lymphocystis disease in Japanese flounder (Paralichthys olivaceus). Aquaculture 2006, 254: 203-210 BUT not Feng et al.

Response 46: We have corrected, and we are sorry for the mistake reference number.

   Herein, the number of reference is [24].

Point 47: L274: Do not start a sentence with ‘And’. Please revise, possibly by consulting a Pivot Sciedit (Certificate No.: pvsc-19100101-3) editor who is a native English-speaker and who has an English language qualification.

Response 47: We have revised it.

   Please see Response 3 in General comments

Point 48: L276: It should be: … and it proved (not provided) that the BMS RM …..

Response 48: we have revised it.

Our results showed the genetic diversity of captive forest musk deer in Barkam center was maintained a high level under closed breeding condition, and it implied that the BMS-RM was effective in inbreeding controlling.

Point 49: L281: In our study we found that the genetic profile of the 1st generation 281 population was the healthiest. This and the next sentence actually suggest that it will be difficult to maintain this high genetic diversity in future generations. With each generation you are losing genetic diversity and in the future this will lead to inbreeding problems and poor health. OK I see, in the following sentence you make that point – GOOD.

Response 49: This is a good suggestion and inevitable result.

The loss of genetic diversity is inevitable in the closed breeding population with the increase of generations in the future, and effective BMS-RM only can reduce the speed of genetic diversity loss and inbreeding increase.

We have improved the discussion.

However, the loss of genetic diversity is inevitable in the closed breeding population with the increase of generations in the future, and the BMS-RM only can reduced the speed of genetic diversity loss and inbreeding increase. Therefore, an effective strategy to control the inbreeding coefficient and to maintain genetic diversity is that new forest musk deer can be introduced into the population, and the individuals exchange among different breeding centers.

Point 50: L288: … but BMS RM rules prevent the introduction of new individuals. Exactly, this is why you should be careful using and proposing the BMS-RM and rather seek for genetic exchange between breeding centres and by introducing fresh blood from the wild.

Response 50: We have deleted the sentence, and strongly proposed to improve the BMS-RM. This is just significance of the study.

Point 51: L289: Do not start a sentence with ‘And’ – this is very poor style! Instead: Moreover, our research also shows

Response 51: Thank you, we have revised it according to your suggestion.

Point 52: L290: Here you say that selective breeding has improved the genetic set-up of the population but in L280 you state that non-random mating and selection lead to a loss of diversity! This is a contradiction – please revise!

Response 52: we have revised the discussion.

We don't think it is contradictory. Captive musk deer population in Barkam center is at the risk of genetic diversity loss judged form the increase of inbreeding coefficient from the 1st generation to the 5th generation, but our result showed that selective breeding also maintained genetic diversity, even improved the genetic structure of the captive musk deer population judged from the average number of alleles and the allele richness.

    Different parameters which judge genetic diversity may reflect the long-term or short-term dynamic changes of genetic diversity.

    However, the loss of genetic diversity is inevitable in the closed breeding population with the increase of generations in the future, and the BMS-RM only can reduced the speed of genetic diversity loss and inbreeding increase. Therefore, an effective strategy to control the inbreeding coefficient and to maintain genetic diversity is that new forest musk deer can be introduced into the population, and the individuals exchange among different breeding centers.

Here’s our new discussion

    2.2 Analysis of genetic diversity in populations at different generations

Protecting the gene pool of a population requires preserving its genetic diversity. DNA replication, non-random mating, selection and drift and other factors all tend to lead to a loss of diversity [38]. In our study we found that the genetic diversity was the highest (HE = 0.742), and Wright’s inbreeding coefficient was the lowest (FIS = 0.001) among the other generations (HE = 0.731, FIS = 0.115). Consistent with our findings, Huang et al. reported that the genetic diversity of the adult captive population was higher than that of younger population [28]. Captive musk deer population in Barkam center is at the risk of genetic diversity loss judged form the increase of inbreeding coefficient from the 1st generation to the 5th generation. .  On the contrary, our result also showed that selective breeding also maintained genetic diversity, even improved the genetic structure of the captive musk deer population judged from the average number of alleles and the allele richness: the average number of alleles increased from 8.25 at the 1st generation to 9.25 at the 5th generation, and the allele richness increased from 6.177 at the 1st generation to 6.325 at the 5th generation. New private alleles appeared at the Mb102C (two new private alleles) and Mb34 (one new private allele)locus,. Conversely, a private allele at Mb116H locus was detected in the 1st generation , but was not detected in the 4th, 3rd and 2nd generation. The loss of private allele in Mb116H locus may be due to the elimination of male individuals in offspring. Huang et al.  came to a similar conclusion in their study of young vs adult captive groups in Miyaluo center [28]. Maybe, different parameters of judge genetic diversity may reflect the long-term or short-term dynamic changes. However, the loss of genetic diversity is inevitable in the closed breeding population with the increase of generations in the future, and the BMS-RM only can reduced the speed of genetic diversity loss and inbreeding increase, or improved genetic structure in short period. Therefore, the effective strategies are to control the inbreeding coefficient and to exchange individuals among different breeding centers for maintaining genetic diversity, and to induce wild forest musk deer for improving genetic diversity.

Point 53: L295 Omit the term ‘population’!! Better say generation groups as you did it in the following line.

Response 53: We have revised it.

    It is worth noting that the newly added genes in generation groups were mostly rare alleles with less than 5% of gene frequency.

Point 54: L297: … locus may be due to the elimination of male individuals in offspring. Wrong grammar – please revise this sentence!

Response 54: We have revised it.

    The loss of private allele in Mb116H locus might be due to that male individuals from offspring didn’t involve in mating.

Point 55: L299: With generation increasing, the number of private alleles decreased. What do you mean by generation increasing. Please revise grammar and phraising!

Response 55: we have improved the discussion, and the sentence “with generation-increasing, the number of private alleles decreased” has been deleted. Please see response 54.

Point 56: L308: Ballou pointed out …..: Make Ballou a proper citation!

Response 56: we have improved the citation.

    Ballou’s study suggested that effectively increase the genetic contribution of new wild founders to a population as well as increase the reproductive life span of existing founders and their close descendents will act to reduce genetic drift and inbreeding effects in the population and thereby minimize the loss of genetic diversity, therefore each of founder should produce at least seven pairs of descendants [32].

Point 57: L311-16: This part is somehow out of context and should be rather placed in the conclusion section. However, I strongly disagree. The only way to maintain a high diversity amongst captice musk deer populations is to exchange genetic material between breeding centres and to occasionally improve diversity by wild caught individuals.

Response 57: We also agree with your opinion: the effective measure is to exchange genetic material between breeding centres in maintaining a high diversity of musk deer populations is and to occasionally improve diversity by wild caught individuals. We aslo put up the opinion in the 3.2 of discussion.

    Therefore, the effective strategies are to control the inbreeding coefficient and to exchange individuals among different breeding centers for maintaining genetic diversity, and to induce wild forest musk deer for improving genetic diversity.

Point 58: L315-16: I wonder whether the main purpose of breeding forest musk deer is to increase genetic diversity or to increase the musk yield for traditional medicine???

Response 58: Not just. We have added the significance of species protection.

Any species on the earth has its significance and value. Captive and artificial breeding are also means of species protection.

Point 59: L321-22: It is difficult to randomly mate in a population with a large number of captive individuals. The solution is a stud book as it is maintained for many other endangered species in captivity. The authors should strongly advocate for the implementation of such a stud book for the whole of China and standardise mating and improve genetic diversity by optimal exchange of breeding males between breeding centres!

Response 59: This is a good suggestion. We also strongly advocate for the implementation of such a stud book for the whole of China and standardize mating and improve genetic diversity by optimal exchange of breeding males between breeding centres

We added the suggestion into the discussion.

     It’s necessary to be further clarified the proper size of N, NBG and NM in order to achieve the minimum inbreeding increment using the BMS-RM under closed breeding condition, and we advocate to exchange individuals among different breeding centers and to occasionally induce wild forest musk deer as the complementary strategy of BMS-RM in China.

Point 60: L334-35: … and t Breeding Management 334 System of Rotated Mating (BMS RM). What is the single ‘t’ for??? Omit!

Response 60: The “t” should be “the” and we have revised it.

Point 61: L337: generation increasing. Better say: with increasing number of generations

Response 61: we have revised it according to the reviewer’s suggestions.

Point 62: L338-40: Therefore, it is necessary to improve the breeding management system or develop new breeding management system f the captive population in the future. How???? You should give some recommendations here of HOW to improve the breeding management system!!

Response 62: We think the conclusion is the direct result of the study and the implication of the result. How to develop a new breeding system has been discussed in our discussion, such as, individuals exchange among different breeding centers and wild forest musk deer induction act as the complementary strategy of BMS-RM.

References

Point 63: Sheng, H. Genus moschus in China. In Wang, S.; Zheng, G.M.; Wang, Q.S. China Red Data Book of Endangered Animals: Pisces. 1998, National Environment Protection Agency of China, Beijing, Agris Publisher. Moschus is a genus and needs to start with a capital letter and should be in italics. Why do you quote the volume on fishes (Pisces)???

Response 63: We are sorry for the mistake, and we have revised the reference.

    [2] Sheng, H. Genus moschus in China. In Wang, S.; Zheng, G.M.; Wang, Q.S. China Red Data Book of Endangered Animals. 1998, National Environment Protection Agency of China, Beijing, Agris Publisher.

Reviewer 2 Report

The authors significantly improved their manuscript following reviewers' comments. However, I have few minor revisions to suggest before publication.

L 23: The 'in' before the words 'Barkam center' was written twice. Please remove one of them. L 35-38: I appreciate that the manuscript was edited by a professional and I can see a general improvement. However, I would suggest to double-check one more time the whole manuscript for a general English editing because there are still few mistakes throughout the text. This sentence (L35-38) provide an example to explain my point: 'is significant lower in the 2nd, 3rd, 4th..' instead of 'is significantly lower than the 2nd, 3rd..' etc.. Using the 'and' too many times throughout the sentence makes the reading difficult to follow and it doesn't sound ok. L 49-50: This is another example: were deer hunted for habitat fragmentation? or do the authors mean that forest musk deer are threatened by habitat fragmentation? Again please do consider to revise your manuscript. L 54-56: I think that the reference provided by the authors doesn't give information on non-invasive technique for musk collection nor about the Chinese breeding program developed in 1958. The reference (Fan, Z.X. et al. The draft genome sequence of forest musk deer (Moschus berezovskii). GigaScience, 2018, 7: 1 7) is about the first genome sequence and gene annotation for the forest musk deer. Please do consider to include an appropriate reference. L 99: Please, modify 'to evaluated' with 'to evaluate'. L100: Please, modify 'and examined' with 'and to examine'. L 105: The section Materials and Methods has to be numbered as 2 while Introduction as n 1 in line. Consequently, Results section will be n 3, Discussion section will be n 4 etc. Please, amend this numeration. L 120: Do the authors mean 'begin to be used for reproduction'? L 126-127: Please consider to edit the sentence 'The breeding of all musk deer in Barkam center is strictly obeyed the rules..' with 'The breeding of all musk deer in Barkam center strictly adheres to the rules..' or similar. L 128: Please, consider to move the sub-chapter '1.2 Blood collection and DNA extraction' below the previous sentence. L 107-128: Response 17 of the authors says that they added a reference for the Breeding Management System. I apologise but I can't see the reference in the text. I found the diagram showed by the authors in their response 17 interesting. It provides a better view of the breeding management system applied in the study. If the authors like, I guess that including a revised version of the diagram may be an extra point added to the paper. L 190-194: This sentence is way too long. Please, consider to edit it. L 201-202: Please, consider to check again the data wrote in these lines. I fully believe that the authors' calculation are correct but what I am discussing about is only related to a switch of info in the text compared to those provided in Table 4. Whereby, if I see Table 4, I read for 'All loci' of the 1st generation that HO is 0.743 and that HE is 0.742. While, in the text the authors wrote the opposite. L 259-261: Please, consider to edit this sentence because too long and not easy to follow. Do the author mean 'possible reasons' not 'impossible reasons'? I may suggest something like: 'Measures of inbreeding control and a larger size of population may help to explain the higher genetic diversity observed in captive forest musk deer compared to their wild counterparts. Etc....' . L 287: Please, remove 'can be' from the sentence. L 299-301: Please, this sentence doesn't sound ok. Consider to change it with something like 'Therefore, it is necessary that the breeding management level of the captive population gradually improves overtime' L 329: Please, remove 'a' after the word' and'. L 327-331: Please, edit this sentence: Do the authors means something like 'Information such as large population with the certain number of individuals (N), the number of individuals within each breeding group (NBG) and the number of breeding groups from the large population should be further clarified in order to achieve the minimum inbreeding increment using the breeding management system under closed breeding condition.' L 334: Please, add 'he' after 't'. L 333-338. I am sorry but this sentence is way too long and confusing to follow. Please, do consider to edit this concept as showed in other examples provided above. L 341-344: According to 'author contribution' the final writing-review and editing was carried out only by author MZ. This looks a bit weird to me because it is important that all authors review and agree on the final version of the manuscript. I am now wondering, did the other authors read-approve the final manuscript?

Author Response

Dear reviewer:

We thank your comments very much, and response all question and suggestion by one-to-one and as follows:

Comments and Suggestions for Authors

The authors significantly improved their manuscript following reviewers' comments. However, I have few minor revisions to suggest before publication.

Point 1: L 23: The 'in' before the words 'Barkam center' was written twice. Please remove one of them.

Response 1: We deleted it

Point 2: L 35-38: I appreciate that the manuscript was edited by a professional and I can see a general improvement. However, I would suggest to double-check one more time the whole manuscript for a general English editing because there are still few mistakes throughout the text.

Response 2: Thank you for the good suggestion. The manuscript has been edited by two editors who are native English-speaker after the manuscript was revised according to the reviewer’s comment. Because language-editor is not experts in genetics, we can't guarantee the accuracy of the use of professional terms.

Point 3: This sentence (L35-38) provide an example to explain my point: 'is significant lower in the 2nd, 3rd, 4th..' instead of 'is significantly lower than the 2nd, 3rd..' etc.. Using the 'and' too many times throughout the sentence makes the reading difficult to follow and it doesn't sound ok.

Response 3: We improved the sentence.

FIS of the 1st generation is significant lower than that of the 2nd to 5th generation (P < 0.01).

Point 4: L 49-50: This is another example: were deer hunted for habitat fragmentation? or do the authors mean that forest musk deer are threatened by habitat fragmentation? Again please do consider to revise your manuscript.

Response 4: We have improved the introduction, and

    The forest musk deer (Moschus berezovskii) is an Asian ungulate that is listed as an endangered species by the International Union for Conservation of Nature (IUCN)[1]. It is included in the CITES Appendix II and is a Level 2 protected wild animal in China [2]. It is widely distributed in many provinces in China and in the northern part of Vietnam [3]. Musk produced by male musk deer has high value, and it is used as a precious natural flavor and an important ingredient of traditional Chinese medicines. Therefore, wild male forest musk deer were illegally hunted for collecting musk. From the late 1960s to the late 1990s, the number of wild musk deer dropped sharply from one million animals to about two hundred thousand [4,5].

Point 5: L 54-56: I think that the reference provided by the authors doesn't give information on non-invasive technique for musk collection nor about the Chinese breeding program developed in 1958. The reference (Fan, Z.X. et al. The draft genome sequence of forest musk deer (Moschus berezovskii). GigaScience, 2018, 7: 1 7) is about the first genome sequence and gene annotation for the forest musk deer. Please do consider to include an appropriate reference.

Response 5: Yes, The reference (Fan Z.X. et al, 2018) didn’t directly given the information on non-invasive technique for musk collection or about the Chinese breeding program developed in 1958, and the references of the article included the reference.

We have recited the reference.

  Sheng, H.; Liu, Z. The Musk Deer in China. Shanghai: The Shanghai Scientific & Technical Publishers; 2007.

Point 6: L 99: Please, modify 'to evaluated' with 'to evaluate'.

Response 6: Thank reviewer, we have revised it.

In this study, our objective was to evaluate genetic structure of the population at the musk deer farm of Sichuan Institute of Musk Deer Breeding in Barkam and examine the change in genetic structure of successive generations based on polymorphisms at twelve microsatellite loci.

Point 7: L100: Please, modify 'and examined' with 'and to examine'.

Response 7: Thank reviewer, we have revised it. Please see Response 6.

Point 8: L 105: The section Materials and Methods has to be numbered as 2 while Introduction as n 1 in line. Consequently, Results section will be n 3, Discussion section will be n 4 etc. Please, amend this numeration.

Response 8: we have added the number.

Point 9: L 120: Do the authors mean 'begin to be used for reproduction'?

Response 9: Yes, we have revised it.

  The offspring of 3-year-old male and 2.5-year-old female musk deer begin to be used for reproduction in the next generation.

Point 10: L 126-127: Please consider to edit the sentence 'The breeding of all musk deer in Barkam center is strictly obeyed the rules..' with 'The breeding of all musk deer in Barkam center strictly adheres to the rules..' or similar.

Response 10: we have revised it according to the reviewer’s suggestion.

Point 11: L 128: Please, consider to move the sub-chapter '1.2 Blood collection and DNA extraction' below the previous sentence.

Response 11: We have moved it to a new line.

  The breeding of all musk deer in Barkam center is strictly adheres to the rules of BMS-RM, and all individuals avoid sib-pair and parent-offspring mating.

Point 12: L 107-128: Response 17 of the authors says that they added a reference for the Breeding Management System. I apologise but I can't see the reference in the text. I found the diagram showed by the authors in their response 17 interesting. It provides a better view of the breeding management system applied in the study. If the authors like, I guess that including a revised version of the diagram may be an extra point added to the paper.

Response 12: We are sorry for missing reference, and we added the reference “[7]”. This is a good suggestion, and we provide the diagram as figure 1.

Point 13: L 190-194: This sentence is way too long. Please, consider to edit it.

Response 13: we have revised it.

    There was a largest number of private alleles (PR = 6) in the 1st generation compared to other subsequent generations, and the number of private alleles continuously decreased from the 1st to 4th generation because some male musk deer offspring didn't participate in the breeding in BMS-RM.The number of private alleles increased from the 4st to 5th generation because rotation mating induced new male musk deer from other shed.

Point 14: L 201-202: Please, consider to check again the data wrote in these lines. I fully believe that the authors' calculation are correct but what I am discussing about is only related to a switch of info in the text compared to those provided in Table 4. Whereby, if I see Table 4, I read for 'All loci' of the 1st generation that HO is 0.743 and that HE is 0.742. While, in the text the authors wrote the opposite.

Response 14: Thank the reviewer for careful checking. We checked the data, and found it is mistake in the text and is right in the table 4.

To confirm the rightness of these data in the manuscript, we check all data in the manuscript again.

Point 15: L 259-261: Please, consider to edit this sentence because too long and not easy to follow. Do the author mean 'possible reasons' not 'impossible reasons'? I may suggest something like: 'Measures of inbreeding control and a larger size of population may help to explain the higher genetic diversity observed in captive forest musk deer compared to their wild counterparts. Etc....' .

Response 15: We sorry for the mistake, and it should be “possible”.

We have revised it.

  The possible reasons is is to take measures of inbreeding control in the population of captive forest musk deer, while the population of wild deer live in fragmentation habitat to lead to a higher degree of inbreeding.

Point 16: L 287: Please, remove 'can be' from the sentence.

Response 16: We have deleted it, and the discussion has been improved.

  Therefore, the effective strategies are to control the inbreeding coefficient and to exchange individuals among different breeding centers for maintaining genetic diversity, and to induce wild forest musk deer for improving genetic diversity.

Point 17: L 299-301: Please, this sentence doesn't sound ok. Consider to change it with something like 'Therefore, it is necessary that the breeding management level of the captive population gradually improves overtime'

Response 17: The discussion has been improved, and discussion section has a big revision.

In discussion of 4.3

However, the loss of genetic diversity is inevitable in the closed breeding population with the increase of generations in the future, and the BMS-RM only can reduced the speed of genetic diversity loss and inbreeding increase, or improved genetic structure in short period. Therefore, the effective strategies are to control the inbreeding coefficient and to exchange individuals among different breeding centers for maintaining genetic diversity, and to induce wild forest musk deer for improving genetic diversity.

It’s necessary to be further clarified the proper size of N, NBG and NM in order to achieve the minimum inbreeding increment using the BMS-RM under closed breeding condition, and we advocate to exchange individuals among different breeding centers and to occasionally induce wild forest musk deer as the complementary strategy of BMS-RM in China.

Point 18: L 329: Please, remove 'a' after the word' and'.

Response 18: We have deleted it, and improved the sentence

  In order to decrease the risk of inbreeding and ΔFIS, the number of individual within the population ( N ), the appropriate number of individuals within  breeding group ( NBG ), the number of breeding groups within the large population (), the number of male musk deer within breeding group ( Nm ) and the exchange of male musk deer across different generations were considered as important factors.

Point 19: L 327-331: Please, edit this sentence: Do the authors means something like 'Information such as large population with the certain number of individuals (N), the number of individuals within each breeding group (NBG) and the number of breeding groups from the large population should be further clarified in order to achieve the minimum inbreeding increment using the breeding management system under closed breeding condition.'

Response 19: we have improved the sentence.

  Please see Response 18 and the following sentence.

It’s necessary to be further clarified the proper size of N, NBG and NM in order to achieve the minimum inbreeding increment using the BMS-RM under closed breeding condition, and we advocate to exchange individuals among different breeding centers and to occasionally induce wild forest musk deer as the complementary strategy of BMS-RM in China.

Point 20: L 334: Please, add 'he' after 't'.

Response 20: We have revised it.

  “and the Breeding Management System of Rotated Mating (BMS-RM) is effective in inbreeding control, but the genetic diversity of population slowly decreased and inbreeding coefficient slowly increased with increasing number of generation, which implied the captive population is facing higher pressure from inbreeding and subsequent loss of genetic diversity.”

Point 21: L 333-338. I am sorry but this sentence is way too long and confusing to follow. Please, do consider to edit this concept as showed in other examples provided above.

Response 21: we have improved the conclusion of the study.

  In summary, the finding of our present study is that the genetic diversity of captive forest musk deer in Barkam center maintained a high level. The Breeding Management System of Rotated Mating (BMS-RM) is effective in maintaining high genetic diversity, but the genetic diversity of population slowly decreased and inbreeding coefficient slowly increased with increasing number of generation, which implied the captive population is facing the pressure of inbreeding and subsequent loss of genetic diversity. Therefore, it is necessary to improve the BMS-RM or develop new breeding management system of the captive population in the future.

Point 22: L 341-344: According to 'author contribution' the final writing-review and editing was carried out only by author MZ. This looks a bit weird to me because it is important that all authors review and agree on the final version of the manuscript. I am now wondering, did the other authors read-approve the final manuscript?

Response 22: Thank your regard, and I confirm the final manuscript was reviewed by all authors. We revised the Conflicts of Interest

This manuscript is a resubmission of an earlier submission. The following is a list of the peer review reports and author responses from that submission.